# EFFECTIVE BIOLOGICAL REPRESENTATION LEARNING BY MASKING GENE EXPRESSION

**Kian Kenyon-Dean**[1]    **Ihab Bendidi**[1,2]    **Alina Selega**[1]    **Jordan M. Sorokin**[1]
**Luca Bertinetto**[1]    **David Errington**[1]    **Oren Kraus**[1]    **Hayley Donnella**[1]
[1]Recursion    [2]Valence Labs
kian.kd@recursion.com   info@rxrx.ai

## ABSTRACT

Transcriptomic foundation models frequently fail to outperform linear baselines despite being trained on massive RNA sequencing corpora exceeding tens of millions of cells. To investigate this, we present TxFM, a transformer masked autoencoder trained on DiverseRNA-1.4M, a novel dataset of 1.4 million bulk and single-cell samples we curated from public data. We demonstrate that data quality can outweigh scale: TxFM outperforms larger models trained on datasets up to 100 times larger. Using previously published benchmarks, we compare TxFM against 16 existing methods and achieve state-of-the-art zero-shot perturbation representation across three held-out cellular contexts and strong performance on single-cell clustering and classification tasks. Ablations show that our architecture enables effective transfer learning by integrating a high masking ratio with a library size-bounded Poisson objective and a rectified tanh activation to enforce output constraints and sparsity. We also show that TxFM's learned gene-specific parameters recover known protein complexes and pathways without supervision. These results establish that curated pretraining and appropriate architectural priors can yield robust transcriptomic representations that generalize across biological contexts.

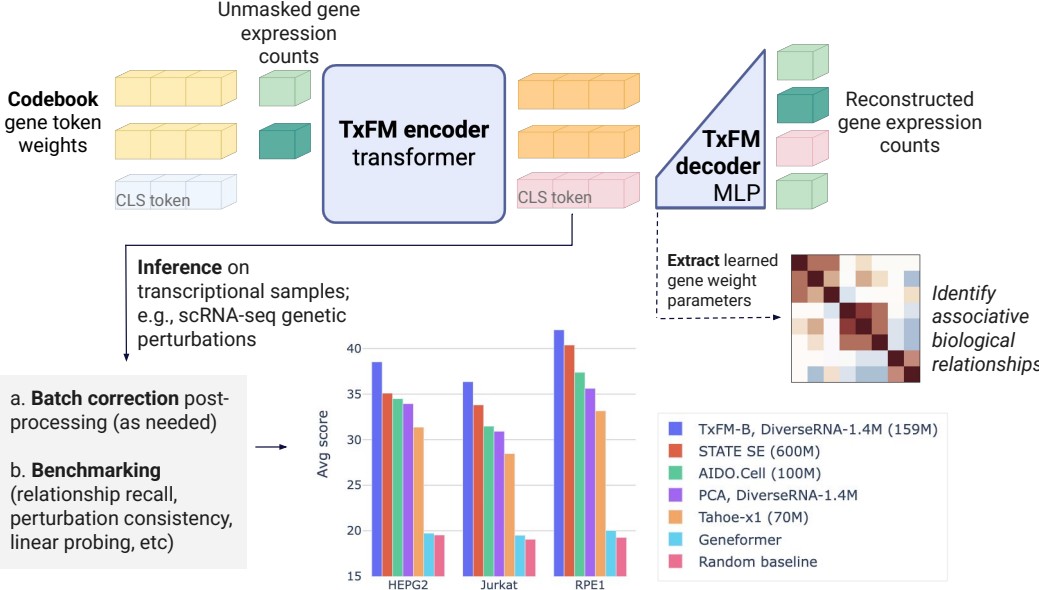

Figure 1: **Overview of our approach.** TxFM is pre-trained for masked reconstruction of gene expression, and evaluated on downstream sample representation quality and gene–gene association structure encoded in its parameters.

# 1 INTRODUCTION

Deep representation learning has transformed vision and language, where foundation models (FMs) pretrained on web-scale data are now standard. In transcriptomics, however, the comparative advantage conferred by bespoke FMs remains equivocal: recent analyses find that many models generalize poorly or fail to outperform simple linear baselines (Szałata et al., 2024; Boiarsky et al., 2024). This suggests challenges in pretraining recipes, motivating comprehensive transfer learning evaluations. Gene expression profiles are high-dimensional, sparse, and noisy, and large public RNA sequencing (RNA-seq) corpora vary widely in quality and biological coverage. A crucial downstream application is *genetic perturbation profiling*, where the expression of many genes is experimentally modified via gene editing technologies such as CRISPR (Lino et al., 2018) and the resulting phenotypes are measured with RNA-seq. Studies like this can shed light on mechanisms of disease and propose novel therapeutic targets Li et al. (2023). To effectively leverage insights from perturbational data and support mechanism discovery, model representations must bridge distinct experimental contexts. Consequently, we prioritize out-of-distribution evaluation on held-out cell contexts to assess transfer capabilities.

We therefore propose TxFM (Figure 1), a self-supervised transformer trained to encode a random subset of gene expression values of a given sample for a lightweight decoder to reconstruct the full expression profile. Our work offers these contributions and insights:

1. **Curation can outweigh scale.** Pretraining on *DiverseRNA-1.4M* – our curated public corpus of 1.4 million high-quality bulk and single-cell RNA-seq samples – often outperforms training on uncurated atlases that are 10 to $100\times$ larger. Our perturbation- and oncology-oriented curation of this public dataset (Table 1) supports accessible representation learning for cancer and drug discovery-relevant settings.

2. **SOTA perturbation representation.** TxFM establishes a new state-of-the-art for representing genetic perturbations across three evaluation datasets with held-out cell lines (Table 2), outperforming 16 public models or FMs and multiple strong baselines. Furthermore, it delivers high aggregate performance across diverse single-cell benchmarks, proving robust generalization across clustering, integration, and classification tasks.

3. **Gene-level interpretability.** TxFM's learned gene parameters (encoder token embeddings and decoder reconstruction weights) recover many known gene-gene relationships (e.g. pathways, protein complexes) without direct supervision.

4. **Practical guidance for effective SSL.** Our extensive ablation study identified the training and architectural requirements (e.g. an extreme 90% masking ratio, Poisson-based loss, and a rectified tanh activation function) that make masked reconstruction an effective transfer learning regime for transcriptomics.

# 2 RELATED WORK

Most existing FMs for transcriptomics use self-supervised learning but differ in approaches to handling the high dimensionality and non-sequential nature of gene expression data. Many models control the input length by employing masking as introduced by scBERT (Yang et al., 2022), scGPT (Cui et al., 2024), and GeneFormer (Theodoris et al., 2023), and often focus on expressed or protein-coding genes only, a practicality that trades off the ability to represent non-coding RNAs which can have important regulatory functions (Statello et al., 2021).

Unlike words in a sentence, genes lack a straight-forward ordering, a challenge that requires architectural adaptation. scGPT adopted custom attention masking with causal self-attention, while GeneFormer used dense attention and ranked sequences by expression. Models also vary in how they represent gene identity and expression. scFoundation (Hao et al., 2024) was the first to embed raw counts instead of binned or ranked values, UCE (Rosen et al., 2023) set the trend of using ESM-2 (Lin et al., 2022) embeddings to represent genes, GenePT (Chen & Zou, 2024) proposed to use ChatGPT-derived embeddings, and scTab (Fischer et al., 2024) used architecture tailored to tabular data. Some FMs jointly model other modalities to better represent biological context: CellPLM (Wen et al., 2023) incorporated spatial transcriptomics and scCello (Yuan et al., 2024) used cell type ontologies. Cell2Sentence (Rizvi et al., 2025) casts gene expression as cell sentences with genes

ordered by decreasing expression for billion-parameter scale models. Long-context AIDO.Cell (Ho et al., 2024), interpretability-focused scPrint (Kalfon et al., 2025), and cross-species Transcript-Former (Pearce et al., 2025) scaled pre-training datasets to 50-100 million cells. Tahoe-100M (Gandhi et al., 2025) recently emerged as the largest small-molecule perturbational dataset to date and multiple recent models pre-trained on it: their companion FM Tahoe-x1, a world-model style GeneJEPA (Litman et al., 2025), and a perturbation prediction FM STATE (Adduri et al., 2025).

However, benchmarking studies found many of these FMs unable to generalize or outperform simple baselines across a range of tasks despite their scale (Boiarsky et al., 2024; Bendidi et al., 2024; Kedzierska et al., 2025; Wong et al., 2025; Ahlmann-Eltze et al., 2025). Indeed, increasing data size does not necessarily translate to improved performance of autoencoders (DenAdel et al., 2024).

**Masked autoencoders (MAEs).** Originally designed for images, MAE (He et al., 2022) offers a self-supervised learning regime with reliable performance and transfer to downstream biological tasks (Kraus et al., 2024; Kenyon-Dean et al., 2025; Kim et al., 2025; Richter et al., 2025). TxFM belongs to the MAE model family but incorporates several adaptations for count-based gene expression data, including an MLP decoder exclusively using the final encoder CLS token (rather than a token-wise transformer decoder), applying reconstruction loss to both masked and unmasked genes, and additional architectural and objective choices that we systematically ablate in § 5.2.

## 3  APPROACH

Let $\mathbf{x} \in \mathbb{N}^G$ be a gene expression count vector for a single-cell (sc) or bulk RNA-seq sample with $G$ profiled genes. We seek low-dimensional embeddings $\mathbf{e} \in \mathbb{R}^d$ ($d \ll G$) that isolate biological signal from high-dimensional sparsity and noise, enabling downstream analyses such as comparing perturbations and inferring gene–gene associations.

We adopt a self-supervised learning framework based on masked reconstruction. TxFM consists of two main components: the **encoder**, a transformer that takes masked gene expression $\mathbf{x}$ as input to produce a sample-level representation $\mathbf{e}$; and the **decoder**, a lightweight MLP that takes $\mathbf{e}$ as input and predicts a full expression vector $\hat{\mathbf{x}} \in \mathbb{R}^G$. Unlike language modeling, where word order is critical, a gene expression profile is an unordered set of measurements taken simultaneously, akin to a bag-of-words. For this reason, we do not model gene expression as a sequence.

**Preprocessing.** Each input gene expression vector $\mathbf{x}$ is normalized to have a total count of $\sum_{i=1}^{G} \mathbf{x}_i = L = 10^5$ (library size normalization), then log-transformed after adding a pseudo-count to accommodate zero counts: $\mathbf{x}_i = \log(\mathbf{x}_i + 1)$, or log1p. This is a standard normalization procedure for RNA-seq analysis (Conesa et al., 2016).

**Masking strategy.** To enable self-supervision, we uniformly at random sample a subset $\mathbf{k} \subseteq \{1, \ldots, G\}$ of size $K$ and only reveal the values $\{\mathbf{x}_i : i \in \mathbf{k}\}$ to the encoder. The remaining gene expression values are masked. Sampling is uniform across all observable genes within each sample; e.g. if $K = 2048$, a cell from the K562 dataset (Replogle et al., 2022) has an effective mask ratio of $\sim75\%$ since there are 8,248 possible genes to unmask, while a sample from GTEX (Table 1) would be masked $\sim94\%$.

**Model.** Each unmasked gene indexed with $i \in \mathbf{k}$ is embedded by concatenating a learnable gene embedding $\mathbf{E}_i \in \mathbb{R}^{d-1}$ with its preprocessed expression value $\mathbf{x}_i$, forming: $\text{token}_i = [\mathbf{E}_i; \mathbf{x}_i] \in \mathbb{R}^d$.

A learnable CLS token is appended to this set and all $K + 1$ tokens are passed through a transformer encoder, relating every unmasked gene to each other through dense self-attention. The transformer's output embedding of the CLS token $\mathbf{e}$ is then passed through an MLP (count) decoder: $\hat{\mathbf{x}} = \phi(\text{MLP}(\mathbf{e}))$, where $\phi$ is a novel *rectified tanh* activation function we introduce and define on gene expression logits $z$ as:

$$\phi(z) = \log(L + 1)\text{ReLU}\left(\tanh\left(\frac{z}{4e}\right)\right). \tag{1}$$

We designed this activation function to respect the non-negative, count-based nature of the data. Unlike pure ReLU, this activation is bounded and asymptotically approaches the normalized upper bound of the data $\log(L + 1)$, preventing divergence by naturally dampening gradients as the predicted count increases (§ C.1, Figure 7).

Table 1: Composition of the default TxFM training dataset after we apply specialized data curation. Each dataset is publicly available. sc: single-cell.

| dataset summary | mode | # samples | # genes |
|---|---|---|---|
| K562 CRISPRi *phenoprints* (Replogle et al., 2022) | sc | 502,080 | 8,248 |
| Glioblastoma (Ruiz-Moreno et al., 2022) | sc | 504,929 | 21,310 |
| MixSeq cancer cell lines (McFarland et al., 2020) | sc | 102,205 | 15,438 |
| sci-Plex compounds (Srivatsan et al., 2020) | sc | 99,300 | 18,486 |
| Gastric metaplasia (Nowicki-Osuch et al., 2023) | sc | 88,399 | 16,445 |
| Tumor microenvironment (Guimarães et al., 2024) | sc | 71,585 | 17,619 |
| Breast cancer (Wu et al., 2021) | sc | 31,542 | 24,712 |
| TCGA (Weinstein et al., 2013) | bulk | 23,733 | 19,594 |
| GTEX (Consortium, 2020) | bulk | 10,526 | 36,695 |
| Total composite dataset: **DiverseRNA-1.4M** | mixed | 1,434,299 | 44,349 |

**Loss function.** TxFM can be trained to minimize any reconstruction loss. By default, we use the following Poisson-based reconstruction loss function on the model's prediction $\hat{\mathbf{x}}$ and the gene expression in the log-normalized training data $\mathbf{x}$ for each sample (derivation in § A.2):

$$\mathcal{L}_{\text{Poisson}}(\mathbf{x}, \hat{\mathbf{x}}) = \frac{1}{G} \sum_{i=1}^{G} (e^{\hat{\mathbf{x}}_i} - \hat{\mathbf{x}}_i e^{\mathbf{x}_i}). \tag{2}$$

By combining this loss with the rectified tanh activation, we constrain the hypothesis class to a library-bounded link function, where the Poisson rate $\lambda \in [1, L+1)$ matches the empirical support of the library-normalized target data with pseudocounts. The rectification in $\phi(z)$ clamps negative gene logits to the minimum rate ($\lambda=1$) without requiring an explicit mixture model of zero-inflation.

**Training dataset.** We train our default model on a curated collection of public transcriptomic datasets with both single-cell and bulk RNA-seq samples, selected with a focus on oncology and perturbational biology. We call this dataset containing $\sim$1.4 million samples *DiverseRNA-1.4M* (Table 1). For the large-scale perturbational dataset of CRISPRi-edited K562 cells (Replogle et al., 2022), we apply a *phenoprint* curation strategy that filters for perturbations with distinct transcriptional profiles (see § B.1) to enrich for biologically meaningful variation; a strategy shown by Kenyon-Dean et al. (2025) to work effectively when training MAEs for microscopy. For the other single-cell datasets, we retain only genes expressed in at least 1,000 cells and cells expressing at least 2,000 genes, yielding an aggregate total of 44,349 unique genes in the training data.

## 4 EXPERIMENTS

To evaluate the biological relevance and generalization capability of FM embeddings, we design experiments that mirror practical scenarios in drug discovery and transcriptomic analysis. Our focus is on tasks that require robust representations across cell contexts unseen during training.

**Zero-shot encoder representation of perturbations.** We adopt the methodology of Bendidi et al. (2024) for benchmarking transcriptomic FMs in zero-shot tasks, assessing five key properties of biological representations: batch effect correction (iLISI), perturbation separation (KNN and linear probing), perturbation consistency, biological knowledge recall, and invertibility to gene count space. We evaluate models across three scRNA-seq datasets of genetic perturbations on cell lines unseen during training: RPE1 (Replogle et al., 2022), HEPG2, and Jurkat (both from Nadig et al. (2024)). Table 2 shows the average results for Jurkat and HEPG2 datasets.[1]

**Cell clustering and classification.** We adopt the benchmark established by Kedzierska et al. (2025) to assess cell type representation through two distinct lenses. First, we evaluate the *geometric structure* of the embedding space using unsupervised clustering ($NMI$, $ARI$) and integration ($ASW_{l/b}$)

---

[1] For TxFM hyperparameters, see § A.1; for details about the evaluation and compared models, § B.1; for extended results for all datasets, § D. We evaluate the best-performing publicly available variant of each model that we benchmarked, see Table 8.

Table 2: **Results on Bendidi et al. (2024) benchmark.** Results on HEPG2 and Jurkat cell lines for CRISPRi genetic perturbations, measuring representation quality in a zero-shot setting (data is unseen during pretraining). Best is in **bold**, second best is underlined. See § D for results for RPE1, score breakdown, std, metric and downstream task descriptions. ilisi: iLISI, lin: linear probing, knn: KNN, p.cst: perturbation consistency, bmdb: biological relationship recall, inv: invertibility to counts, avg: average of all scores.

| model | Nadig HEPG2 | | | | | | | Nadig Jurkat | | | | | | |
|---|---|---|---|---|---|---|---|---|---|---|---|---|---|---|
| | ilisi↑ | lin↑ | knn↑ | p.cst↑ | bmdb↑ | inv↑ | avg↑ | ilisi↑ | lin↑ | knn↑ | p.cst↑ | bmdb↑ | inv↑ | avg↑ |
| **Count data baselines** | | | | | | | | | | | | | | |
| Random label shuffle | 0.70 | 0.66 | 5.45 | 0.00 | 12.95 | 28.14 | *19.68* | 0.70 | 0.54 | 7.46 | 0.04 | 11.59 | 24.77 | *19.20* |
| Raw counts | 0.60 | 23.21 | 9.75 | 2.60 | 32.15 | 56.38 | *30.81* | 0.63 | 20.47 | 10.36 | 11.55 | 35.25 | 57.28 | *33.06* |
| (Lib+Log)Norm | 0.65 | 21.51 | 11.93 | 10.49 | 35.32 | **60.09** | *34.12* | 0.62 | 16.73 | 11.07 | 14.69 | 34.71 | **60.50** | *33.37* |
| (Lib+Log)Norm+5k HVG | 0.67 | 19.22 | 11.38 | 8.09 | 34.75 | 50.54 | *31.88* | 0.64 | 13.07 | 10.76 | 20.95 | 37.29 | 50.61 | *32.94* |
| (Lib+Log)Norm+1024HVG | 0.69 | 15.04 | 10.74 | 24.13 | 44.06 | 44.46 | *34.65* | 0.68 | 9.67 | 10.38 | 16.95 | 38.68 | 39.44 | *30.67* |
| **ChatGPT gene embeddings** | | | | | | | | | | | | | | |
| GenePT Large | **0.73** | 18.65 | 8.28 | 24.78 | 42.01 | 40.91 | *34.77* | **0.74** | 14.30 | 8.58 | 24.47 | 38.11 | 35.74 | *32.55* |
| **Pretrained FMs** | | | | | | | | | | | | | | |
| Geneformer [CellXGene] | 0.71 | 2.21 | 5.44 | 0.04 | 11.30 | 28.41 | *19.89* | 0.71 | 2.97 | 7.44 | 0.00 | 10.60 | 25.00 | *19.63* |
| GeneJEPA [Tahoe-100M] | 0.70 | 2.64 | 5.74 | 0.41 | 23.68 | 38.86 | *23.64* | 0.70 | 3.14 | 7.59 | 0.04 | 14.96 | 33.56 | *21.66* |
| scTab [CellXGene] | 0.70 | 4.59 | 6.67 | 1.93 | 32.89 | 38.80 | *25.91* | 0.70 | 3.70 | 7.72 | 0.12 | 21.12 | 32.79 | *22.69* |
| CellPLM | 0.71 | 5.12 | 7.52 | 3.68 | 23.03 | 40.68 | *25.25* | 0.71 | 4.27 | 8.19 | 1.79 | 23.99 | 34.81 | *24.06* |
| UCE [CellXGene] | 0.70 | 5.23 | 7.20 | 6.31 | 31.32 | 40.47 | *26.92* | 0.71 | 4.97 | 7.97 | 2.95 | 28.61 | 34.87 | *25.07* |
| scCello [CellXGene] | 0.71 | 8.50 | 7.44 | 6.95 | 36.77 | 40.14 | *28.50* | 0.70 | 7.26 | 8.69 | 0.37 | 28.45 | 34.77 | *25.09* |
| scPrint-L [CellXGene] | 0.71 | 2.96 | 5.93 | 0.16 | 26.89 | 35.57 | *23.80* | 0.70 | 8.43 | 9.24 | 5.13 | 33.34 | 35.48 | *27.04* |
| scGPT [CellXGene] | 0.70 | 7.21 | 8.51 | 11.46 | 36.69 | 41.30 | *29.35* | 0.71 | 7.66 | 8.61 | 5.53 | 32.45 | 35.70 | *26.86* |
| Tahoe-x1 [Tahoe-100M] | 0.70 | 9.53 | 9.83 | 15.39 | 39.30 | 43.25 | *31.36* | 0.71 | 9.61 | 9.40 | 10.15 | 34.02 | 36.37 | *28.47* |
| TranscriptFormer-Sapiens [CellXGene] | 0.70 | 12.16 | 8.94 | 19.33 | 35.14 | 42.77 | *31.55* | 0.70 | 10.77 | 8.68 | 20.40 | 31.38 | 37.30 | *29.92* |
| AIDO.Cell-100M [CellXGene] | 0.71 | 16.98 | 10.26 | 25.05 | 41.42 | 42.28 | *34.52* | 0.71 | 15.07 | 9.46 | 19.54 | 37.08 | 36.69 | *31.53* |
| Cell2Sentence | 0.70 | 16.12 | 9.99 | 24.52 | 42.24 | 43.44 | *34.49* | 0.71 | 13.93 | 9.39 | 26.21 | 39.11 | 37.21 | *32.82* |
| TxFM-B [TF-Sapiens data] (ours) | 0.70 | 17.81 | 10.59 | 25.28 | 43.31 | 43.87 | *35.30* | 0.71 | 16.52 | 9.82 | 22.59 | 37.97 | 39.77 | *32.98* |
| STATE-SE [CxG + Tahoe-100M + scBC] | 0.69 | 17.20 | 10.79 | 23.13 | 43.94 | 45.15 | *35.10* | 0.69 | 17.15 | 9.92 | 24.45 | 39.44 | 42.65 | *33.81* |
| **Fit on our public train set (Zero-shot)** | | | | | | | | | | | | | | |
| PCA [DiverseRNA-1.4M] | 0.69 | 11.11 | 11.50 | 20.53 | 43.34 | 48.21 | *34.00* | 0.69 | 7.60 | 10.04 | 16.83 | 37.44 | 44.56 | *30.91* |
| scVI [DiverseRNA-1.4M] | 0.70 | 14.16 | 10.83 | 17.28 | 40.00 | 43.17 | *32.69* | 0.71 | 12.97 | 10.43 | 16.18 | 36.16 | 37.36 | *30.68* |
| TxFM-S [DiverseRNA-1.4M] (ours) | 0.70 | 23.08 | 13.11 | 29.52 | **45.88** | 44.56 | *37.82* | 0.70 | 22.18 | 11.60 | 28.51 | **42.56** | 38.89 | *35.75* |
| TxFM-B [DiverseRNA-1.4M] (ours) | 0.70 | **25.93** | **13.14** | **32.47** | 45.26 | 44.40 | *38.63* | 0.69 | **23.79** | **11.59** | **31.27** | 42.08 | 40.42 | *36.52* |

metrics, which measure local density and batch invariance. Second, we evaluate the *informational content* of the embeddings using supervised classification probing. By fitting linear and 2-layer MLP classifiers, we test whether cell identity is linearly separable or easily recoverable, regardless of the unsupervised clustering geometry. See Table 4 for results and § B.2 for detailed protocols.

**Intrinsic evaluation of model parameters.** TxFM learns gene representations in its encoder tokens and decoder reconstruction parameters because genes are both features in transcriptomic data and the targets for our SSL objective. We investigate these representations with gene-gene relationship recall metrics (Celik et al., 2024; Kraus et al., 2024) as a transfer learning evaluation of TxFM variants in our ablations in order to determine a performant model architecture (Table 5).

## 5 RESULTS

Table 2 benchmarks the quality of TxFM's perturbation representations against 16 FMs and strong baselines. We first observe that simple count data baselines establish a surprisingly high bar for zero-shot representation: standard pre-processing (library size normalization with log-transformation, Lib+Log) or selecting highly variable genes (HVG) frequently outperform existing FMs. Against this strong backdrop, TxFM-B (trained on DiverseRNA-1.4M) emerges as the only model to consistently surpass the baselines, achieving the highest overall score averaged across all tasks on each held-out dataset. It outperforms all competing FMs, as well as baselines like scVI (Lopez et al., 2018) and PCA also trained on DiverseRNA-1.4M. TxFM-B outperforms the best alternative model, STATE-SE (Adduri et al., 2025), despite having nearly $4\times$ fewer parameters and being trained on $100\times$ less data.

Two key comparisons isolate the drivers of this performance. First, *data quality can outweigh volume:* TxFM-B trained on the curated DiverseRNA-1.4M outperforms the same architecture trained on the TF-Sapiens atlas, which contains $\sim$57 million (mostly uncurated) cells. Next, *architecture is*

Table 4: **Cell type representation performance averaged across the five datasets from the Kedzierska et al. (2025) benchmark.** We evaluate clustering ($NMI$, $ARI$, $ASW$), batch integration ($ASW_{l/b}$), and classification probing accuracy. $\diamond$ and $\star$ denote, respectively, simple count baseline (no training) and models pretrained without test distribution overlap (zero-shot). For each metric, the best performance is in **bold**, second best is underlined. The rank column expresses the average rank of a method across metrics and is used to sort methods from best to worst. Table 7 shows per-dataset results.

| | | Clustering | | | Batch Int. | Classification | | | |
| | | | | | | Linear Probe | | 2-layer MLP | |
| Rank↓ | model | ASW ↑ | NMI ↑ | ARI ↑ | $ASW_{l/b}$ ↑ | Acc ↑ | F1 ↑ | Acc ↑ | F1 ↑ |
|---|---|---|---|---|---|---|---|---|---|
| 3.1 | TxFM-B [TF-Sapiens data] | $53.9_{\pm0.2}$ | $65.8_{\pm0.6}$ | $46.6_{\pm1.8}$ | $82.4_{\pm0.6}$ | 85.6 | **71.6** | 86.5 | **72.7** |
| 4.4 | STATE-SE | $50.9_{\pm0.1}$ | $63.9_{\pm0.6}$ | $41.9_{\pm1.3}$ | **$95.4_{\pm0.2}$** | 85.1 | 70.5 | 85.9 | 71.5 |
| 4.6 | scGPT | $53.5_{\pm0.2}$ | $65.5_{\pm0.6}$ | $50.9_{\pm1.6}$ | $87.0_{\pm0.6}$ | 84.0 | 68.5 | 83.9 | 69.0 |
| 4.7 | TxFM-B [DiverseRNA-1.4M] $\star$ | **$54.4_{\pm0.1}$** | $64.2_{\pm0.7}$ | $46.0_{\pm0.9}$ | $85.7_{\pm0.6}$ | 84.7 | 68.3 | 85.3 | 69.0 |
| 5.0 | TxFM-S [DiverseRNA-1.4M] $\star$ | $54.5_{\pm0.2}$ | $65.1_{\pm0.5}$ | $46.2_{\pm1.1}$ | $85.6_{\pm0.6}$ | 84.6 | 67.4 | 85.1 | 68.8 |
| 6.0 | AIDO.Cell | $47.6_{\pm0.3}$ | $58.2_{\pm0.8}$ | $36.3_{\pm1.9}$ | $80.3_{\pm0.6}$ | **86.5** | 71.5 | **86.7** | 72.1 |
| 6.2 | scTab | $53.6_{\pm0.2}$ | **$68.3_{\pm0.5}$** | **$57.2_{\pm1.6}$** | $91.1_{\pm0.4}$ | 80.8 | 59.0 | 81.7 | 62.2 |
| 6.2 | Transcriptformer [TF-Sapiens data] | $52.3_{\pm0.1}$ | $66.0_{\pm0.8}$ | $47.3_{\pm3.1}$ | $82.8_{\pm0.7}$ | 83.0 | 67.1 | 82.8 | 66.3 |
| 7.8 | (Lib+Log)Norm+2000HVG $\diamond$ | $50.4_{\pm0.2}$ | $62.8_{\pm0.5}$ | $42.9_{\pm0.9}$ | $93.6_{\pm0.2}$ | 79.1 | 62.6 | 81.5 | 67.3 |
| 7.9 | PCA [DiverseRNA-1.4M] $\star$ | $50.0_{\pm0.1}$ | $55.3_{\pm0.9}$ | $36.6_{\pm1.8}$ | $91.2_{\pm0.4}$ | 82.8 | 66.1 | 83.9 | 67.4 |
| 9.9 | scVI [DiverseRNA-1.4M] $\star$ | $50.4_{\pm0.2}$ | $56.8_{\pm0.5}$ | $37.5_{\pm1.4}$ | $80.5_{\pm0.9}$ | 78.6 | 61.4 | 80.1 | 64.7 |

*decisive:* when controlled for training data, TxFM-B (trained on TF-Sapiens) outperforms the $2\times$ larger TranscriptFormer model trained on the same corpus.

**SSL fine-tuning results.** We evaluate an in-domain adaptation setting where TxFM is SSL fine-tuned on the target dataset itself (Table 3) with the same masked reconstruction objective (§A.1.1), reflecting a common practitioner work-flow to obtain a biologically meaningful low-dimensional representation when generalization beyond the target dataset is not the primary goal. In this setting, fine-tuning of TxFM-B pre-trained on DiverseRNA-1.4M outperforms PCA and scVI trained on the same target dataset, yielding an aggregate 5 to 19% relative improvement in average performance over PCA and scVI.

Table 3: Average perturbation representation score (Bendidi et al., 2024) for PCA, scVI, and SSL-fine-tuned TxFM-B models trained on RPE1, HEPG2, Jurkat datasets separately.

| model / data | RPE1 | HEPG2 | Jurkat |
|---|---|---|---|
| PCA | 42.64 | 38.63 | 34.89 |
| scVI | 37.65 | 35.63 | 32.62 |
| TxFM-B (ours) | **44.85** | **40.78** | **38.01** |

## 5.1 Cell type representation benchmarking

Table 4 presents an aggregate assessment of model performance, revealing a tension between geometric and informational objectives. While unsupervised clustering rewards models that compress cells into dense, local neighborhoods, this can sometimes come at the cost of global separability. Our results show that a representation can appear geometrically scattered (lower clustering scores) while still encoding robust, linearly separable cell identities (high probing accuracy), necessitating a holistic view of performance.

**Aggregate performance and zero-shot significance.** Navigating this trade-off, TxFM-B trained on TF-Sapiens emerges as the most capable generalist, delivering consistent top-tier performance across both clustering and classification tasks. This highlights the scaling efficiency of our architecture: when trained on large-scale atlases, TxFM-B outperforms STATE-SE Adduri et al. (2025), a model with nearly $4\times$ the learnable parameters trained on a dataset roughly $3\times$ larger. It is crucial to contextualize these results within the training distributions. Models trained on large atlases can benefit from partial test leakage, as evaluation datasets like Tabula Sapiens (Jones et al., 2022) are frequently subsets of their self-supervised training corpora. In contrast, the $\star$ marked versions of TxFM operate in a strict zero-shot setting, yet perform near the top of the ranking. This underscores that representations learned from curated perturbational data can generalize very effectively.

**Task-specific inductive biases.** Some models show performance skewed toward specific metrics, revealing distinct design trade-offs. scTab Fischer et al. (2024) excels at clustering. Its TabNet backbone Arik & Pfister (2021) uses sequential attention which, similarly to decision trees, can cre-

ate sharp partitions ideal for discrete clustering metrics (ASW/NMI/ARI). However, this can result in a fragmented geometry which is not conducive to strong classification. AIDO.Cell (Ho et al., 2024), conversely, excels at classification but struggles with clustering, suggesting its embeddings capture rich signal but distribute it with high intra-class variance (e.g., retaining library size noise). Supervised probes can learn to ignore this noise, but unsupervised clustering algorithms cannot.

Ultimately, no single model dominates every metric, but TxFM offers an effective compromise: retaining the local geometry needed for clustering and the invariance required for batch correction, while ensuring cell types remain extractable for downstream classification.

## 5.2 ABLATION OF THE MAIN PROPERTIES OF TxFM

Table 5 presents our ablation study (inspired by He et al. (2022)) to probe our model architecture choices for TxFM. We evaluate TxFM as an encoder of perturbational scRNA-seq data by measuring perturbation consistency across three datasets unseen during training (summarized in the column "perts"). We also evaluate the quality of the non-perturbational gene features captured within TxFM's learned parameters by measuring whole-genome gene-gene relationship recall of the (enc)oder's input layer gene tokens and (dec)oder's gene reconstruction weights.[2]

(a) **Reconstruction loss.** Each loss is defined with respect to the log1p-transformed ground truth counts $\mathbf{x}$ and the reconstructed count predictions $\hat{\mathbf{x}}$. We implemented a loss function based on the negative binomial distribution, which required additional learned dispersion parameters per gene (§ A.2). Our Poisson-based loss function (Equation 2) empirically yields significant improvements over the others on each evaluation.

(b) **Count preprocessing.** We confirm that the standard practice of normalizing count data by library size (Conesa et al., 2016) is important for achieving good performance with TxFM. This is similar to sample-specific pixel normalization / self-standardization used to preprocess images for MAEs (He et al., 2022; Kraus et al., 2024).

(c) **Mask ratio.** TxFM's effective masking ratio is dataset-dependent, given the different numbers of sequenced genes in each dataset (Table 1). Unmasking 2048 genes on DiverseRNA-1.4M proves to be effective, yielding an average mask ratio of ~90%. This regime outperforms unmasking 4096 genes, despite having significantly less training compute. Our high mask ratio stands in contrast to previous work, which used much lower values (Theodoris et al., 2023; Rosen et al., 2023; Cui et al., 2024). We also differ from AIDO.Cell (Ho et al., 2024) as they pass [MASK] tokens for each gene through the encoder during training, an inefficiency proven harmful to downstream performance when training MAEs (He et al., 2022).

(d) **Masking strategy.** To account for scRNA-seq sparsity, we test frequency-weighted masking based on how often each gene is nonzero in the training set. Let $m_i$ be the number of training samples where gene $i$ has nonzero expression, and define sampling weights $w_i \propto m_i^\tau$ for a temperature $\tau$. Positive $\tau$ preferentially unmasks frequently expressed genes, while negative $\tau$ favors rarely expressed genes. In practice, uniform masking ($\tau = 0$) was simplest and gave the best overall trade-off, so we use it by default.

(e) **Decoder depth.** We assess decoder complexity by comparing a linear mapping to varying-depth MLPs with residual connections. Results indicate minimal sensitivity to depth, confirming that the encoder captures a sufficiently rich representation in the CLS token to enable effective reconstruction with a simple decoder.

(f) **Decoder count activation.** Our proposed *rectified tanh* activation (Equation 1) significantly improves zero-shot perturbation consistency compared to three alternatives.

(g) **Encoder backbone.** We ablate encoder capacity by training Small (-S), Base (-B), and Large (-L) variants on DiverseRNA-1.4M. Results indicate a performance plateau on this dataset: moving

---

[2]If not otherwise specified, TxFM's settings are: library size normalization followed by log1p count preprocessing, $K = 2048$ unmasked genes, uniform random mask sampling, a 4-layer MLP decoder, rectified tanh count activation function (Equation 1), Poisson reconstruction loss (Equation 2), Base transformer architecture, trained on DiverseRNA-1.4M (Table 1) for 200 epochs, approaching a mean of 1,000 H100 GPU hours per training run. Other hyperparameters are described in § A.1, Table 6. See § C for additional details.

Table 5: **TxFM architecture ablation experiments**. We report total perturbation consistency (p. cst) on RPE1, Jurkat, and HEPG2 combined (perts) and whole-genome gene-gene relationship recall (Kraus et al., 2024) captured in a TxFM's learned parameters: (enc)oder gene tokens and (dec)oder gene weights. Default TxFM architecture is marked in gray. Performance is significantly different from Default when indicated with $\star$ if $p_{adj} < 0.05$ (§ C). Best performance in an ablation is in **bold**.

| loss | perts | enc | dec |
|---|---|---|---|
| MSE | 27.2$^\star$ | 31.1$^\star$ | 41.8$^\star$ |
| SmoothL1 | 23.4$^\star$ | 23.2$^\star$ | 38.1$^\star$ |
| Poisson | **37.3** | **32.6** | **43.9** |
| Neg. Bin. | 33.9$^\star$ | 29.4$^\star$ | 41.2$^\star$ |

(a) **Reconstruction loss.** Our Poisson-based loss function significantly improves performance.

| preprocessing | perts | enc | dec |
|---|---|---|---|
| log1p | 27.9$^\star$ | 25.7$^\star$ | 42.2$^\star$ |
| LibNorm, log1p | **37.3** | **32.6** | **43.9** |

(b) **Count preprocessing.** Normalizing count data for library size helps to learn high-quality perturbational and non-perturbational gene representations.

| # unmasked | perts | enc | dec |
|---|---|---|---|
| 512 | 31.9$^\star$ | 26.8$^\star$ | 40.2$^\star$ |
| 1024 | 35.2$^\star$ | 31.9$^\star$ | **44.3** |
| 2048 | **37.3** | **32.6** | 43.9 |
| 4096 | 35.6$^\star$ | 25.3$^\star$ | 42.8$^\star$ |

(c) **Mask ratio.** Training with 2048 unmasked gene tokens (∼90% mask ratio) yields strong performance despite 4096 having more training compute.

| temperature $\tau$ | perts | enc | dec |
|---|---|---|---|
| -1 | 31.3$^\star$ | 32.5 | 42.7$^\star$ |
| -0.5 | 34.4$^\star$ | **34.3$^\star$** | 43.6 |
| 0 (uniform) | 37.3 | 32.6 | **43.9** |
| 0.5 | **37.6** | 28.9$^\star$ | 42.0$^\star$ |
| 1.0 | 36.4 | 25.7$^\star$ | 40.3$^\star$ |

(d) **Masking strategy.** Masking genes uniformly at random offers a good performance trade-off vs frequency-weighted masking.

| decoder | perts | enc | dec |
|---|---|---|---|
| linear | 36.5 | 32.3 | 43.6 |
| 1-layer MLP | 36.0 | 32.3 | 43.8 |
| 4-layer MLP | **37.3** | **32.6** | **43.9** |
| 8-layer MLP | 36.5 | 32.0$^\star$ | 43.6 |

(e) **Decoder depth.** A linear or MLP decoder works effectively with minimal significant differences.

| activation | perts | enc | dec |
|---|---|---|---|
| ReLU (diverges) | 34.4$^\star$ | 26.7$^\star$ | 43.9 |
| clamped ReLU | 34.6$^\star$ | 32.4 | **44.3** |
| rect. tanh | **37.3** | **32.6** | 43.9 |
| softmax | 27.6$^\star$ | 16.2$^\star$ | 15.3$^\star$ |

(f) **Decoder count activation.** Our rectified tanh activation function prevents divergence and improves representation quality.

| backbone | perts | enc | dec |
|---|---|---|---|
| -S (57M) | 33.2$^\star$ | 29.2$^\star$ | 43.9 |
| -B (159M) | **37.3** | **32.6** | **43.9** |
| -L (403M) | 37.2 | 28.2$^\star$ | 43.1$^\star$ |

(g) **Encoder backbone.** Scaling TxFM to Base architecture on DiverseRNA-1.4M performed best on these tasks.

| train data (# samples) [compute] | perts | enc | dec |
|---|---|---|---|
| DiRNA w/o K562 cells (932K) | 31.4$^\star$ | 31.5$^\star$ | 42.0$^\star$ |
| DiRNA + K562 controls (1.0M) | 34.3$^\star$ | 32.0$^\star$ | 42.2$^\star$ |
| DiRNA + curated K562 (1.4M) | **37.3** | 32.6 | 43.9 |
| DiRNA + full K562 (2.8M) | 35.6$^\star$ | 31.2$^\star$ | **44.9$^\star$** |
| DiRNA + full K562 (2.8M) [2x] | 36.7 | 29.8$^\star$ | 44.7$^\star$ |
| TF-Sapiens data (57M) [4x] | 30.3$^\star$ | **35.2$^\star$** | 40.3$^\star$ |

(h) **Dataset curation.** A phenoprints-oriented data curation strategy of DiverseRNA-1.4M is an effective way to train TxFM-B.

from -S to -B improves both perturbation and gene token representations, but further scaling to -L does not yield significant gains despite achieving lower validation reconstruction loss (§ A.1.2).

**(h) Dataset curation.** We ablate our curation strategy for the DiverseRNA-1.4M dataset by varying the inclusion of its largest component: the Replogle et al. (2022) K562 CRISPRi dataset (∼1.9M cells). Evaluating five distinct strategies (§ C.2), we find that including K562 data consistently improves performance, even when restricted to only 72,000 control (no perturbation) cells. Our default *phenoprint* curation strategy, which prunes ∼75% of the cells from the K562 dataset, results in improved perturbation representations, while trading off signal between encoder gene tokens and decoder gene weights. Conversely, scaling to the $40\times$ larger TF-Sapiens dataset (with $4\times$ compute) degrades perturbation representations and decoder structure, despite improving token embeddings.

## 5.3 Epochwise and layerwise analysis of TxFM

TxFM learns gene representations in both its encoder token embeddings and the decoder reconstruction matrix; cosine similarities between these gene representations recover known biological

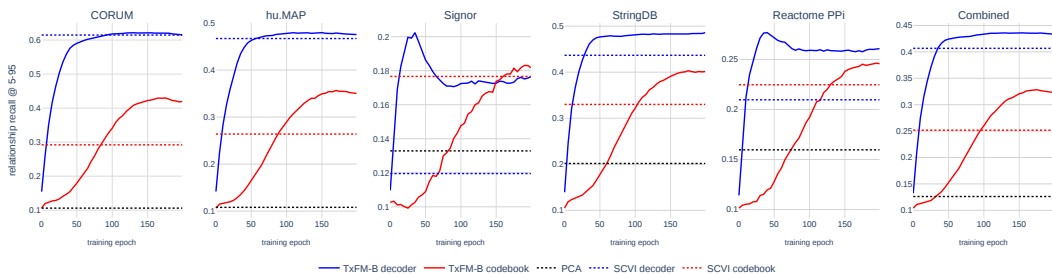

Figure 2: **Epochwise analysis of TxFM during training.** Gene-gene relationship recall (Kraus et al., 2024) performance on 5 databases (plus all combined) of gene representations extracted from the Default TxFM-B codebook and decoder, as a function of training epoch. Dashed lines indicate the recall for the equivalent layers of the scVI and PCA weight matrix trained on DiverseRNA-1.4M.

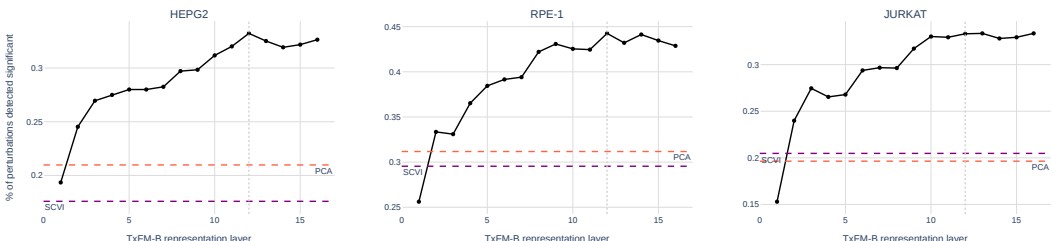

Figure 3: **Layerwise inference-time analysis of TxFM.** Perturbation consistency on HEPG2, RPE-1, and Jurkat datasets for each layer of our Default TxFM-B (encoder layers followed by decoder layers) compared to scVI and PCA (each trained on DiverseRNA-1.4M).

relationships (§ B.3). Figure 2 tracks relationship recall during training of TxFM-B, compared to scVI and PCA trained on the same data. Decoder-based relationships emerge early, while codebook relationships improve more gradually. On Signor and Reactome (Perfetto et al., 2016; Fabregat et al., 2018), decoder recall peaks in the first quarter of training and then declines. Interestingly, Signor is the only database where the final epoch's codebook exceeds the decoder, a trend also seen for scVI. TxFM's learned gene weights outperform those of scVI, indicating potential for novel biological relationship discovery for the purpose of finding new drug targets (Schenone et al., 2013).

Figure 3 evaluates aggregated perturbation (CLS) representations for three unseen cell lines, extracted from each encoder layer and decoder MLP layer. Representation quality improves after the first encoder layer and continues to increase through the 12th (default) layer, with little change in the decoder MLP. TxFM-B's perturbation representations outperform those of scVI and PCA and the best layer is the final transformer block, unlike some large vision MAEs which find optimal performance at intermediate blocks (Alkin et al., 2024; Kenyon-Dean et al., 2025).

## 6 CONCLUSION

We presented TxFM, a data-efficient self-supervised masked autoencoder optimized for the high-dimensional, sparse nature of transcriptomic data. Our results demonstrate that intentional data curation allows TxFM to outperform larger models trained on much larger uncurated cell atlases. This suggests that, in specialized scientific domains, data composition is a more potent driver for effective foundation model transfer learning than sheer volume. From an ML architecture standpoint, we empirically isolate our extreme masking strategy (90%), rectified tanh activation, and Poisson-based loss as the primary drivers of strong transfer learning performance. TxFM establishes a new state-of-the-art for genetic perturbation representation, while learning high-recall gene association maps in its parameters without supervision. Altogether, our results demonstrate TxFM as a practical transcriptomics foundation model with robust transfer across biological contexts.

**Limitations and reproducibility.** TxFM was trained and evaluated exclusively on freely accessible public data. While our oncology-oriented curation improves perturbation representation, distribu-

tional biases may limit generalization to other contexts. Additional research is required to determine scaling laws on transcriptomics data. We release the novel modeling components and benchmarking tools here: `https://github.com/recursionpharma/opentxfm`.

**Acknowledgements.** This work was funded and supported by Recursion and Recursion employees.

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

APPENDIX CONTENTS

## A  TxFM TRAINING

### A.1  TxFM TRAINING HYPERPARAMETERS

All TxFM models were trained with data-distributed parallel methods over H100 GPUs on a large-scale compute cluster. Table 6 describes relevant training hyperparameters. Each model used a one-cycle cosine learning rate decay schedule with 10% warm-up using the AdamW optimizer with max learning rate 1e-3, betas (0.9, 0.999), epsilon 1e-6, weight decay of 1 divided by the learning rate times the total number of training steps, bfloat16-mixed precision, with additional standard techniques such as LayerScale. The models with 1.4M samples were trained on our curated DiverseRNA-1.4M (Table 1). The TxFM-B trained on 57M samples used the much larger Cellx-Gene Transcriptformer-Sapiens dataset (Pearce et al., 2025); preliminary investigations indicated that $K = 1024$ was more suitable for the sparser scRNA-seq data in this larger dataset vs the denser DiverseRNA-1.4M data. All other model parameter settings, if not otherwise stated, follow the TxFM-B Default model as per the ablations in § 5.2.

Table 6: Training hyperparameters for TxFM variations.

| hyperparameter | TxFM-S | TxFM-B Default | TxFM-B CxG | TxFM-L |
|---|---|---|---|---|
| training dataset # samples | 1.4M | 1.4M | 57M | 1.4M |
| training epochs | 200 | 200 | 50 | 200 |
| K (# unmasked tokens) | 2048 | 2048 | 1024 | 2048 |
| global batch size | 1536 | 1536 | 6144 | 1536 |
| total training GPU-hours | 400 | 864 | 4096 | 2500 |
| # encoder blocks | 6 | 12 | 12 | 24 |
| # MHSA heads | 6 | 12 | 12 | 16 |
| model embedding dim. | 384 | 768 | 768 | 1024 |
| stochastic depth rate | 0.1 | 0.1 | 0.1 | 0.3 |
| total learnable parameters | 57M | 159M | 185M | 403M |

#### A.1.1  SSL-FINETUNING

In Table 3 we evaluate the impact of training unsupervised models directly on the evaluation data to address biological use cases pertaining to individual dataset analysis, where PCA is typically used as the representation. (This is in contrast to the zero-shot setting in Table 2, where the evaluation

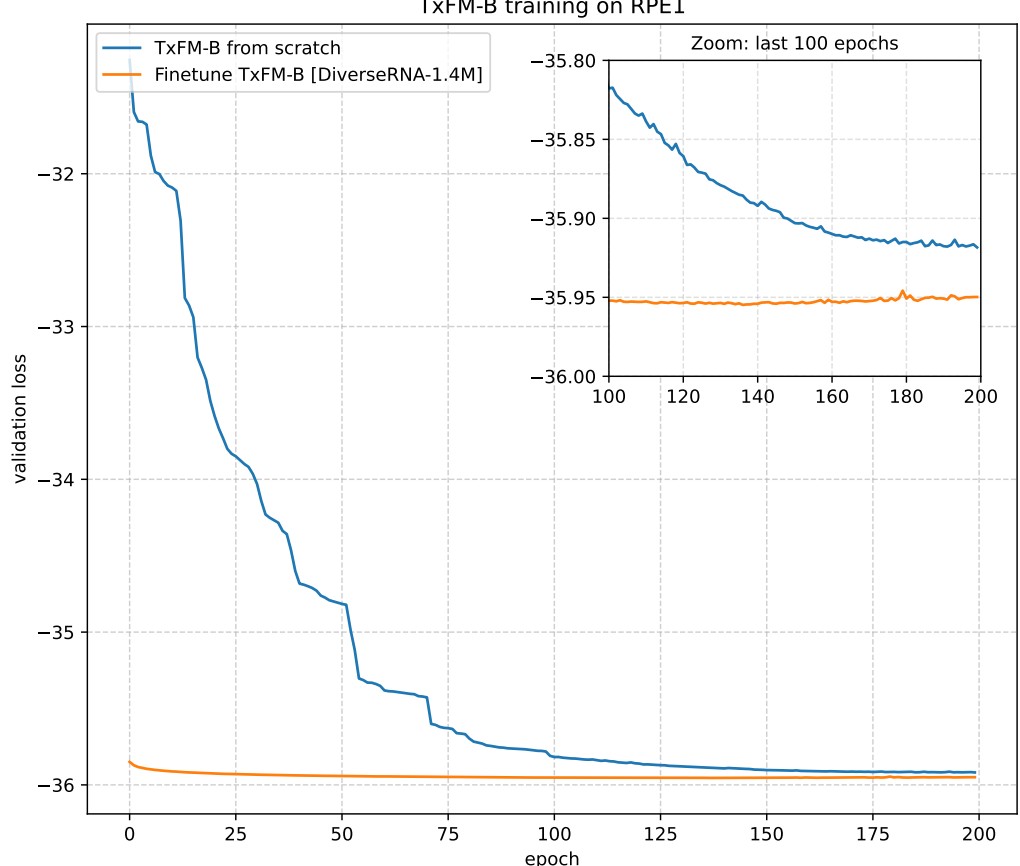

Figure 4: Example of SSL training a TxFM-B on the RPE1 dataset from scratch versus SSL fine-tuning TxFM-B Default (pretrained on DiverseRNA-1.4M) on the RPE1 dataset.

data is entirely unseen during unsupervised training and where we sought to determine how strong pretrained SSL models generalize to new cellular contexts.) Therefore, we evaluated performance when training PCA and scVI models directly on each of the evaluation datasets (HEPG2, Jurkat, RPE1) and compared to self-supervised finetuning the default TxFM-B on those datasets. Our SSL finetuning performs masked reconstruction on the evaluation dataset with the same setup as pretraining (2048 unmasked genes, 200 epochs). In Figure 4 we visualize the difference in our Poisson reconstruction loss between training a TxFM-B from scratch on RPE1 versus finetuning. We observe that the pretrained model is already very close to high performance on RPE1, and the benefit of finetuning is made clear as it obtains better generalization that the from-scratch TxFM, as the validation set contains held out experimental batches from the assay.

### A.1.2    SCALING ON DIVERSERNA-1.4M

Figure 5 shows training loss curves for the three different TxFM backbones trained on DiverseRNA-1.4M (Table 5 (g), training hyperparameters in Table 6). As we can see, TxFM-L obtains the best reconstruction loss on the validation set during training. However, this unfortunately did not translate to better downstream transfer learning performance. It is possible that the parameter count for TxFM-B saturates DiverseRNA-1.4M, and therefore more curated data should be obtained to effectively scale to TxFM-L. While we would have liked to explore doing so in this work, we sought to instead prioritize our compute budget to comprehensive architecture ablations on the TxFM-B backbone, which already required considerable training compute per standard run. Future work should explore scaling laws and data curation further.

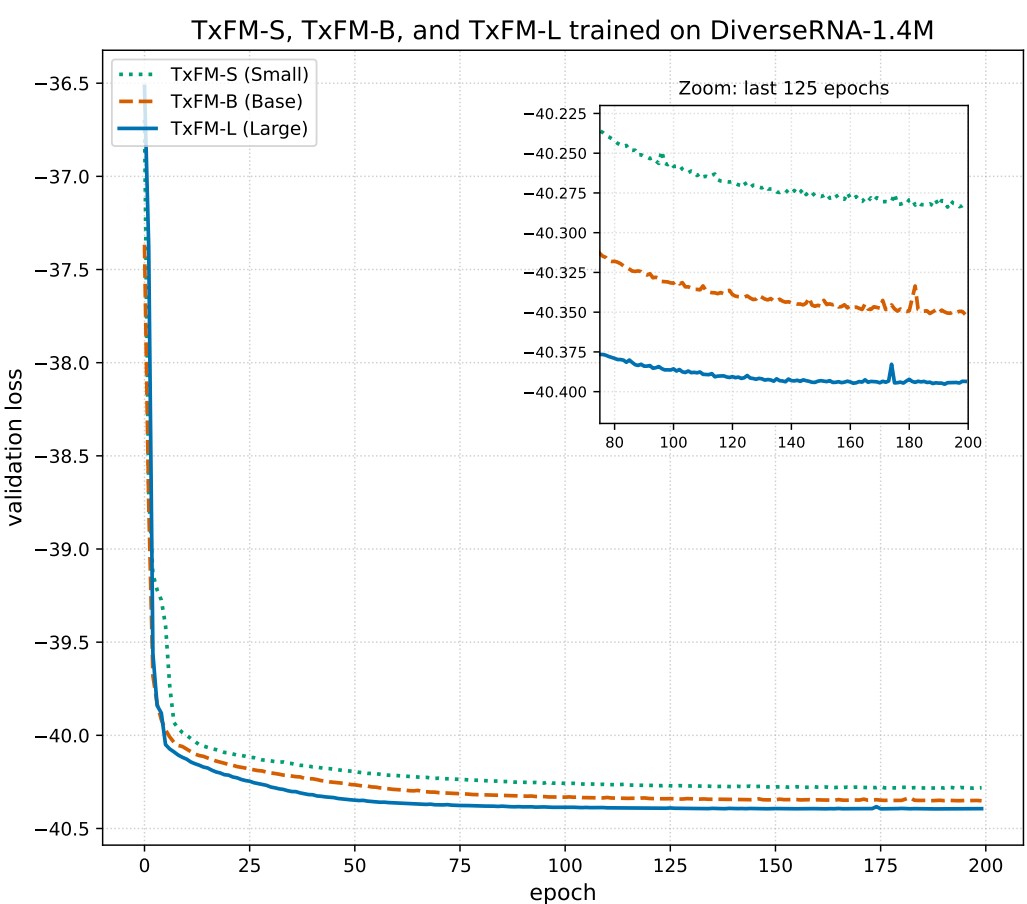

Figure 5: Validation set (held out experimental batches from each of the composite datasets) reconstruction loss curves for the three different TxFM backbones evaluated in Table 5 (g).

## A.2 Loss functions

**Preprocessing.** We preprocess raw gene expression counts by first performing library size normalization. Library size is the total count summed across all genes in a sample. As library size varies between e.g. single cells, it is a common step to normalize all cells to have equal library size $L$ so that individual gene counts are comparable to each other. In this work, we set $L = 10^5$. We then log-transform normalized gene counts, with a prior addition of a pseudocount. This is another common step in RNA-seq analysis, which enables log-transform in the presence of zero gene counts (which are very often found in scRNA-seq data).

$$\tilde{x}_{ng} = \frac{x_{ng}}{\sum_{i=1}^{G} x_{ni}} \cdot L \tag{3}$$

$$\tilde{x}_{ng} = \log(\tilde{x}_{ng} + 1) \tag{4}$$

$x_{ng}$ is the raw gene expression count for gene $g$ in sample $n$, $G$ is the total number of genes profiled in that sample, and $\log$ is the natural logarithm.

**Poisson-based loss.** For an expression count $x_{ng}$, the Poisson negative log-likelihood is defined as follows, while we drop the constant below:

$$-\log p(x_{ng}|\lambda) = -\log\left(\frac{\lambda^{x_{ng}} e^{-\lambda}}{x_{ng}!}\right)$$
$$= -x_{ng} \log \lambda + \lambda \tag{5}$$

Setting the learned rate parameter to our model's prediction as $\hat{\mathbf{x}}_g = \log \lambda$ for the target log-count $\mathbf{x}_g$ (preprocessed as explained above), we recover the loss:

$$\mathcal{L}_{\text{Poisson}}(\mathbf{x}_g, \hat{\mathbf{x}}_g) = e^{\hat{\mathbf{x}}_g} - \hat{\mathbf{x}}_g \cdot e^{\mathbf{x}_g} \tag{6}$$

We model genes in a sample as i.i.d. and thus sum loss values over genes to obtain $\mathcal{L}_{\text{Poisson}}(\mathbf{x}, \hat{\mathbf{x}})$ for an input gene expression vector $\mathbf{x}$.

**Gradient of the loss.** Putting together our rectified tanh activation function $\phi$ and the Poisson-based loss, we can write our loss for a gene count $\mathbf{x}_i$ and its model-generated logit $z$ as:

$$\mathcal{L} = e^{\phi(z)} - \phi(z) e^{\mathbf{x}_i} \tag{7}$$

Taking the gradient, we obtain:

$$\frac{\delta \mathcal{L}}{\delta z} = e^{\phi(z)} \phi'(z) - e^{\mathbf{x}_i} \phi'(z) \tag{8}$$

$$= \phi'(z) \left(e^{\phi(z)} - e^{\mathbf{x}_i}\right) \tag{9}$$

The derivative of the rectified tanh (§ C.1) is given by:

$$\frac{\delta \phi}{\delta z} = \frac{\delta}{\delta z} \left(\log(L+1)\text{ReLU}\left(\tanh\left(\frac{z}{4e}\right)\right)\right) \tag{10}$$

$$= \begin{cases} \frac{\log(L+1)}{4e} \text{sech}^2\left(\frac{z}{4e}\right) & z > 0 \\ 0 & z \leq 0 \end{cases} \tag{11}$$

Putting this together and setting $\alpha = \log(L+1), \beta = 4e$ for brevity, we obtain the gradient of the loss, where the hyperbolic secant function acts as a saturating gate on the Poisson error term:

$$\frac{\delta \mathcal{L}}{\delta z} = \begin{cases} \frac{\alpha}{\beta} \text{sech}^2\left(\frac{z}{\beta}\right) \left(e^{\alpha \tanh\left(\frac{z}{\beta}\right)} - e^{\mathbf{x}_i}\right) & z > 0 \\ 0 & z \leq 0 \end{cases} \tag{12}$$

**Negative binomial-based loss.**    For the ablation experiments, we implemented a negative binomial loss function using the Pytorch `distributions` package. We followed the parameterization by Risso et al. (2018) that models the mean and inverse dispersion as parameters of the distribution and allowed gene-specific inverse dispersion parameters $\theta_g$ shared across samples. The distribution is then defined as follows for a gene count $x_{ng}$:

$$p(x_{ng}|\mu_{ng},\theta_g) =$$
$$\binom{x_{ng} + \theta_g - 1}{x_{ng}} \left(\frac{\theta_g}{\theta_g + \mu_{ng}}\right)^{\theta_g} \left(\frac{\mu_{ng}}{\theta_g + \mu_{ng}}\right)^{x_{ng}} \tag{13}$$

We scaled the learned mean $\mu_{ng}$ for each gene count by the cell size factor defined as $s_n = \frac{\sum_g x_{ng}}{L}$. This implies the following reparameterization: $r = \theta_g$ for the number of failures $r$ and $p = \frac{s_n \mu_{ng}}{s_n \mu_{ng} + \theta_g}$ for the probability of success $p$. The loss for a given gene count is given by the negative log-likelihood of $\text{NB}(r, p)$ and loss values are again summed over genes for a given sample.

For the Poisson loss, we use library size-normalized counts with pseudocounts as targets as it maintains the same loss minima as using raw integer counts. For the negative binomial loss, we used the same data preprocessing for consistency. Training TxFM with the negative binomial loss on raw counts without library size normalization yielded similar results to the Neg. Bin. model as reported in Table 5, with slightly lower perts and enc metrics, slighly higher dec metrics, and the same significance levels.

## B    EVALUATION METHODOLOGY

### B.1    PERTURBATION REPRESENTATION BENCHMARKS

#### B.1.1    EVALUATION SETUP

We adopt the hierarchical evaluation framework introduced in Bendidi et al. (2024), which assesses model performance on gene perturbation tasks using a biologically motivated metric suite. Each metric evaluates a different aspect of model utility, with aggregation occurring across runs, metrics, and tasks: results of evaluations performed at 5 different seeds are first averaged together, then groups of metrics constituting a downstream task are averaged together for one final metric per downstream task. We separately apply different post-processing techniques to the embeddings and count baselines before evaluation: centering on negative controls, standardization, and TVN (Ando et al., 2017). For each downstream task and evaluated model pair, we pick the post-processing technique resulting in the best performing average score.

**Batch Effect Correction (ilisi):**    To quantify batch mixing in the latent space, we use the Integration Local Inverse Simpson's Index (iLISI). A higher iLISI indicates better mixing of batches and hence more effective batch effect reduction. The final iLISI score for each model is averaged across all batches.

**Linear Separability (lin):**    This metric assesses how linearly distinguishable different perturbations are in the latent space using a linear classifier trained on known perturbation labels. Top-1 and Top-5 accuracy using unseen batches for evaluation are reported after being averaged together.

**Latent Space Organization (knn):**    Local structure is assessed via k-nearest neighbor retrieval of perturbation-matched samples across disjoint biological batches. Accuracy is reported for an average of Top-1 and Top-5 retrieval.

**Perturbation Consistency (p.cst):**    This metric measures whether embeddings of the same perturbation are more similar to each other than expected by chance. We compute perturbation consistency by comparing a perturbation's similarity to a null distribution. Here, we refer to all cells that received a given gene knockout (regardless of the guide used) when we say "perturbation". To assemble a null distribution of cosine similarities, we sample $N$ random cells and compute cosine similarity between their embeddings and the mean embedding of each gene perturbation (computed

across corresponding cells). We remove pairwise similarities that correspond to the random cell's perturbation to exclude self-self similarities. In our experiments, we set $N = 10,000$. Then, for each perturbation, we compute mean leave-one-out cosine similarity across the cell embeddings. Finally, for each perturbation, we compute empirical p-value by comparing its similarity to the null distribution. Perturbations with $p < 0.01$ are considered significantly consistent. The final score is the fraction of consistent perturbations.

Perturbation consistency is computed post-alignment consisting of a PCA transform fit on the non-targeting control samples and followed by standardizing the transformed embeddings within each batch.

**Biological Relationship Recall (bmdb):** To test zero-shot biological reasoning, cosine similarities between perturbation embeddings are used to predict gene-gene relationships. Predicted links (top 5% most similar/dissimilar pairs) are compared to curated databases (e.g., CORUM, HuMAP). Recall is computed per database and averaged across databases.

**Latent Space Interpretability (inv):** This is composed of an average of two metrics: Spearman correlation and Structural Integrity. Spearman correlation measures how well latent embeddings can be linearly decoded back into gene expression values. This tests biological interpretability of learned representations. Structural Integrity assesses how well perturbation-induced gene expression changes are preserved in reconstructed profiles. For each batch, we compute the Frobenius norm of the difference between the negative control-centered predicted and the negative control-centered actual gene expression matrices:

$$\text{Structural Distance} = \frac{1}{B} \sum_{b=1}^{B} \frac{1}{n_b} \left\| \tilde{Y}_{\text{pred}}^{(b)} - \tilde{Y}_{\text{actual}}^{(b)} \right\|_F$$

where $B$ is the number of batches, $n_b$ the number of samples in batch $b$, and $\tilde{Y}$ denotes gene expression centered by the batch-specific control. The Structural Integrity is then defined as:

$$\text{Structural Integrity} = 1 - \frac{\text{Structural Distance}}{\text{Structural Distance}_{\text{max}}}$$

where Structural Distance$_{\text{max}}$ is an upper bound estimated from the actual data. Higher scores indicate better preservation of perturbation-induced gene expression structure.

### B.1.2 BENCHMARKED APPROACHES

We evaluate TxFM across a range of baselines including: commonly used existing methods (PCA and scVI (Lopez et al., 2018)); simple transformations applied directly to the test data (raw counts, library size + log normalization, and its variants with 5000 or 1024 highly variable genes); a baseline of 1024 highly variable genes with randomly shuffled labels (Random label shuffle); models using ChatGPT-derived embeddings (GenePT Large, Chen & Zou (2024)); and large pretrained models trained on CellxGene or similar datasets (Geneformer (Theodoris et al., 2023), CellPLM (Wen et al., 2023), scTab (Fischer et al., 2024), UCE (Rosen et al., 2023), scCello (Yuan et al., 2024), scPrint-L (Kalfon et al., 2025), TranscriptFormer-Sapiens (Pearce et al., 2025), scGPT (Cui et al., 2024), AIDO.Cell-100M (Ho et al., 2024)), GeneJEPA (Litman et al., 2025), Tahoe-x1 (Gandhi et al., 2025), Cell2Sentence (Rizvi et al., 2025), STATE (Adduri et al., 2025).

We report results in two settings: (i) zero-shot, where models are trained on data disjoint from the test set and (ii) fit-on-evaluation-data, where models are trained or fine-tuned on the test set. Raw counts baseline uses unprocessed gene counts, Lib+Log applies library size normalization followed by a log-transform, HVG variants subset the genes to either 5000 or 1024 highly variable ones, and Random label shuffle keeps gene inputs fixed while randomizing the perturbation labels. PCA and scVI are evaluated in both settings using the standard benchmark splits.

Our TxFM models include TxFM-B DiverseRNA-1.4M, trained on a curated 1.4 million sample dataset, and TxFM-B TF-Sapiens, trained on a larger ~50 million sample dataset. These models al-

low to isolate the effects of data curation versus training architecture and enable a direct comparison with FMs trained on the same data such as TranscriptFormer-Sapiens.

For the zero-shot setting, we trained multiple scVI models on DiverseRNA-1.4M with the following parameters: latent dimensionality of 768, a single hidden layer with 1024 units, AdamW optimizer, lr=0.0005, three variants of likelihoods (negative binomial, zero-inflated negative binomial, Poisson), two settings for weight decay (default value of 1e-06 or 0.03), and other settings set to default values. For the Poisson model, we adjusted the learning rate to 0.0001 and used default weight decay to help with training stability. We compared these scVI models for zero-shot performance on the evaluation data using gene-gene relationship recall and perturbation consistency benchmarks (described in § B.3 and under *Perturbation Consistency* above). The model trained with the negative binomial likelihood and default weight decay performed best on both benchmarks for all three datasets; we used this model to report scVI zero-shot performance and evaluate its codebook and decoder representations. For the fit-on-evaluation-data setting, we trained scVI on each dataset following the procedure used in Bendidi et al. (2024): using 8000 highly variable genes, latent dimensionality of 256, a single hidden layer with 512 units, no dropout, a zero-inflated negative binomial likelihood and gene-level dispersion parameters.

For Cell2Sentence (Rizvi et al., 2025), we use the 2 billion parameter pretrained version and tried several different prompting strategies. We found that their default recommended prompting strategy performed better than prompting it to predict the perturbation, arriving at the following template:

"The following is a list of $N$ gene names ordered by descending expression level in a Homo sapiens cell. Your task is to give the cell type which this cell belongs to based on its gene expression. Cell sentence: $g_1, g_2, \ldots, g_N$. The cell type corresponding to these genes is:",

such that $N$ is the number of genes (sorted by expression value in a given sample) we include in the prompt. The sample-level embedding taken is the last token of the model's hidden state of the sequence. We found that $N = 1000$ seemed to perform best in perturbation consistency on the HEPG2 data (i.e., $N = 200$ yielded 23.3% p. cst, $N = 500$ yielded 24.7%, $N = 1000$ yielded 24.9%, but $N = 2000$ yielded 23.5%).

## B.2 CELL TYPE CLUSTERING AND CLASSIFICATION BENCHMARK

We evaluate embeddings on clustering and batch integration tasks on the five single-cell datasets[3] used in Kedzierska et al. (2025). Additionally, we assess representation quality through classification probing using both linear classifiers and 2-layer MLPs trained on the learned embeddings. For clustering and batch integration, we employ metrics introduced by Luecken et al. (2022) and implemented in the `scib` package[4].

### B.2.1 CLUSTERING TASK

The clustering task assesses whether cell embeddings preserve meaningful biological structure by measuring how well cells cluster according to their true cell types. We employ Louvain clustering with resolution optimization: we test 10 different resolution parameters and select the resolution that maximizes NMI between discovered clusters and ground truth labels. Results are obtained by subsampling 10,000 cells with 10 random seeds and reporting the average. Three complementary metrics evaluate different aspects of biological preservation:

**Normalized Mutual Information** (NMI)**:** Measures information overlap between discovered clusters $\mathcal{C}$ and true cell type labels $\mathcal{L}$:

$$\text{NMI}(\mathcal{C}, \mathcal{L}) = \frac{2 \cdot I(\mathcal{C}, \mathcal{L})}{H(\mathcal{C}) + H(\mathcal{L})} \tag{14}$$

where $I(\mathcal{C}, \mathcal{L}) = \sum_{c,\ell} p(c, \ell) \log \frac{p(c,\ell)}{p(c)p(\ell)}$ is mutual information and $H(\cdot)$ denotes entropy. High NMI indicates that knowing cluster assignments provides substantial information about true cell types.

---

[3]We note that the *Pancreas* dataset contains non-integer values, implying prior normalization. As the original raw counts are not available and we expect negligible impact on performance, these values were used as input for all models.

[4]`https://scib.readthedocs.io/`

Table 7: Per-dataset results for the Kedzierska et al. (2025) benchmarks in Table 4.

| | Clustering | | | Batch Int. | Classification | | | |
| | | | | | Linear Probe | | 2-layer MLP | |
| | ASW ↑ | NMI ↑ | ARI ↑ | ASW$_{l/b}$ ↑ | Acc ↑ | F1 ↑ | Acc ↑ | F1 ↑ |
|---|---|---|---|---|---|---|---|---|
| **Pan-immune** | | | | | | | | |
| (Lib+Log)Norm+2000HVG | 50.7 ± 0.1 | 58.0 ± 0.6 | 33.2 ± 1.0 | 92.0 ± 0.2 | 86.0 | 78.6 | 87.2 | 80.3 |
| PCA [DiverseRNA-1.4M] | 50.4 ± 0.1 | 53.4 ± 0.4 | 30.5 ± 1.5 | 93.5 ± 0.2 | 82.9 | 75.1 | 84.3 | 79.2 |
| scGPT | 52.7 ± 0.1 | 62.8 ± 0.5 | 40.6 ± 0.8 | 88.3 ± 0.2 | 85.1 | 81.7 | 85.9 | 83.0 |
| AIDO.Cell | 49.5 ± 0.1 | 56.8 ± 0.5 | 32.0 ± 0.9 | 82.6 ± 0.4 | 90.3 | 86.4 | 90.7 | 87.6 |
| scTab | 58.5 ± 0.2 | 83.6 ± 0.7 | 76.8 ± 1.7 | 88.4 ± 0.4 | 89.9 | 80.2 | 90.4 | 81.1 |
| scVI [DiverseRNA-1.4M] | 50.5 ± 0.2 | 53.5 ± 0.4 | 29.7 ± 1.7 | 84.7 ± 0.4 | 81.9 | 78.0 | 83.6 | 81.1 |
| STATE | 50.8 ± 0.1 | 60.4 ± 0.5 | 34.2 ± 1.3 | 96.4 ± 0.2 | 88.1 | 81.4 | 88.6 | 83.3 |
| Transcriptformer | 52.6 ± 0.1 | 72.0 ± 0.5 | 55.1 ± 1.6 | 84.9 ± 0.4 | 90.0 | 86.3 | 90.3 | 86.8 |
| TxFM-B [TF-Sapiens data] | 55.7 ± 0.2 | 66.0 ± 0.5 | 36.5 ± 0.9 | 81.4 ± 0.3 | 90.2 | 86.7 | 90.7 | 87.6 |
| TxFM-S [DiverseRNA-1.4M] | 52.9 ± 0.2 | 61.7 ± 0.7 | 36.1 ± 1.9 | 84.5 ± 0.3 | 87.4 | 83.0 | 88.6 | 85.1 |
| TxFM-B [DiverseRNA-1.4M] | 53.1 ± 0.1 | 62.0 ± 0.6 | 35.8 ± 1.6 | 86.3 ± 0.4 | 88.3 | 83.9 | 89.4 | 85.9 |
| **Pancreas** | | | | | | | | |
| (Lib+Log)Norm+2000HVG | 55.4 ± 0.0 | 75.0 ± 0.4 | 56.0 ± 0.4 | 93.0 ± 0.1 | 71.9 | 60.8 | 75.4 | 70.4 |
| PCA [DiverseRNA-1.4M] | 49.8 ± 0.0 | 39.9 ± 0.6 | 17.4 ± 2.0 | 83.7 ± 0.6 | 95.0 | 85.7 | 94.8 | 83.8 |
| scGPT | 50.9 ± 0.2 | 55.3 ± 0.6 | 36.3 ± 1.9 | 79.9 ± 1.4 | 90.3 | 75.8 | 87.8 | 77.8 |
| AIDO.Cell | 45.3 ± 0.3 | 48.4 ± 0.8 | 22.3 ± 1.8 | 71.7 ± 0.5 | 95.6 | 83.2 | 93.8 | 77.7 |
| scTab | 51.5 ± 0.1 | 54.4 ± 0.8 | 50.2 ± 4.8 | 90.6 ± 0.2 | 75.2 | 50.0 | 78.5 | 60.0 |
| scVI [DiverseRNA-1.4M] | 45.6 ± 0.2 | 41.6 ± 0.2 | 17.2 ± 2.0 | 70.3 ± 1.5 | 73.2 | 61.0 | 76.2 | 67.6 |
| STATE | 51.6 ± 0.0 | 60.7 ± 0.5 | 26.8 ± 0.8 | 93.4 ± 0.2 | 93.4 | 82.4 | 94.3 | 82.6 |
| Transcriptformer | 48.1 ± 0.2 | 56.0 ± 0.5 | 34.3 ± 1.5 | 73.5 ± 0.8 | 75.7 | 57.2 | 73.7 | 52.4 |
| TxFM-B [TF-Sapiens data] | 51.8 ± 0.1 | 50.9 ± 0.4 | 25.4 ± 2.5 | 75.2 ± 1.1 | 89.4 | 82.2 | 89.9 | 83.2 |
| TxFM-S [DiverseRNA-1.4M] | 51.1 ± 0.2 | 50.6 ± 0.3 | 22.8 ± 1.2 | 80.0 ± 0.9 | 90.9 | 71.4 | 91.3 | 72.5 |
| TxFM-B [DiverseRNA-1.4M] | 51.1 ± 0.2 | 50.6 ± 0.3 | 22.8 ± 1.2 | 80.0 ± 0.9 | 90.3 | 73.0 | 90.7 | 72.6 |
| **PBMC 12k** | | | | | | | | |
| (Lib+Log)Norm+2000HVG | 51.4 ± 0.0 | 69.1 ± 0.4 | 59.9 ± 0.7 | 99.3 ± 0.0 | 94.8 | 90.9 | 95.5 | 92.2 |
| PCA [DiverseRNA-1.4M] | 51.6 ± 0.0 | 68.4 ± 0.7 | 61.4 ± 3.6 | 99.0 ± 0.0 | 93.7 | 87.4 | 94.4 | 88.4 |
| scGPT | 62.9 ± 0.1 | 83.8 ± 0.8 | 85.8 ± 3.7 | 96.8 ± 0.2 | 97.2 | 95.9 | 97.1 | 95.8 |
| AIDO.Cell | 58.7 ± 0.1 | 76.7 ± 0.9 | 65.1 ± 1.6 | 93.1 ± 0.2 | 97.5 | 96.2 | 97.5 | 96.6 |
| scTab | 57.8 ± 0.0 | 79.4 ± 0.4 | 82.5 ± 0.6 | 98.3 ± 0.1 | 95.0 | 90.2 | 95.1 | 91.7 |
| scVI [DiverseRNA-1.4M] | 59.0 ± 0.1 | 74.4 ± 0.5 | 66.8 ± 2.0 | 96.0 ± 0.1 | 95.5 | 91.0 | 96.1 | 92.3 |
| STATE | 51.2 ± 0.1 | 74.4 ± 0.7 | 65.0 ± 1.3 | 99.2 ± 0.1 | 95.7 | 92.2 | 96.0 | 92.3 |
| Transcriptformer | 61.0 ± 0.1 | 79.8 ± 2.1 | 74.5 ± 9.6 | 94.3 ± 0.2 | 97.2 | 96.1 | 97.4 | 96.5 |
| TxFM-B [TF-Sapiens data] | 59.3 ± 0.0 | 86.6 ± 0.9 | 89.1 ± 2.5 | 96.5 ± 0.1 | 97.5 | 96.2 | 97.5 | 96.2 |
| TxFM-S [DiverseRNA-1.4M] | 66.0 ± 0.0 | 87.7 ± 0.7 | 92.3 ± 0.6 | 96.0 ± 0.2 | 97.1 | 95.1 | 97.2 | 95.0 |
| TxFM-B [DiverseRNA-1.4M] | 66.0 ± 0.0 | 87.7 ± 0.7 | 92.3 ± 0.6 | 96.0 ± 0.2 | 97.1 | 94.2 | 97.1 | 95.1 |
| **PBMC 68k** | | | | | | | | |
| (Lib+Log)Norm+2000HVG | 48.5 ± 0.1 | 46.1 ± 0.6 | 27.5 ± 1.5 | 96.9 ± 0.2 | 78.2 | 68.9 | 82.2 | 73.0 |
| PCA [DiverseRNA-1.4M] | 49.8 ± 0.1 | 43.8 ± 2.3 | 21.4 ± 0.4 | 96.5 ± 0.5 | 65.7 | 55.0 | 67.0 | 54.6 |
| scGPT | 51.0 ± 0.2 | 50.5 ± 0.6 | 32.8 ± 0.9 | 91.6 ± 0.7 | 70.3 | 58.7 | 70.3 | 57.1 |
| AIDO.Cell | 47.7 ± 0.2 | 44.0 ± 1.2 | 23.5 ± 2.9 | 88.3 ± 0.7 | 72.6 | 61.6 | 73.1 | 61.9 |
| scTab | 50.2 ± 0.1 | 49.0 ± 0.4 | 19.9 ± 0.5 | 94.2 ± 0.6 | 68.5 | 56.7 | 69.0 | 56.8 |
| scVI [DiverseRNA-1.4M] | 49.9 ± 0.2 | 41.9 ± 1.1 | 22.2 ± 0.4 | 91.2 ± 0.9 | 68.3 | 56.2 | 68.4 | 57.1 |
| STATE | 50.1 ± 0.0 | 45.8 ± 1.1 | 24.4 ± 1.7 | 98.6 ± 0.2 | 68.0 | 58.3 | 69.0 | 59.3 |
| Transcriptformer | 49.9 ± 0.2 | 49.0 ± 0.4 | 21.6 ± 1.6 | 90.1 ± 1.0 | 73.5 | 62.8 | 72.8 | 61.6 |
| TxFM-B [TF-Sapiens data] | 50.1 ± 0.2 | 49.8 ± 0.5 | 28.4 ± 1.3 | 89.7 ± 0.8 | 74.5 | 64.2 | 74.6 | 63.2 |
| TxFM-S [DiverseRNA-1.4M] | 50.5 ± 0.1 | 50.1 ± 0.6 | 24.5 ± 0.6 | 91.2 ± 0.8 | 70.4 | 58.3 | 70.7 | 59.0 |
| TxFM-B [DiverseRNA-1.4M] | 50.0 ± 0.2 | 46.0 ± 1.3 | 23.4 ± 0.5 | 92.6 ± 0.6 | 69.1 | 58.3 | 69.8 | 57.5 |
| **Tabula Sapiens** | | | | | | | | |
| (Lib+Log)Norm+2000HVG | 46.1 ± 1.0 | 65.7 ± 0.5 | 37.8 ± 0.9 | 86.7 ± 0.6 | 64.7 | 13.7 | 67.1 | 20.4 |
| PCA [DiverseRNA-1.4M] | 48.4 ± 0.2 | 70.9 ± 0.4 | 52.1 ± 1.1 | 83.2 ± 0.5 | 76.6 | 27.2 | 79.1 | 30.9 |
| scGPT | 50.2 ± 0.3 | 75.1 ± 0.4 | 58.8 ± 0.7 | 78.3 ± 0.7 | 77.2 | 30.5 | 78.3 | 31.4 |
| AIDO.Cell | 36.8 ± 0.5 | 65.0 ± 0.5 | 38.4 ± 1.9 | 65.7 ± 1.1 | 76.4 | 30.0 | 78.2 | 36.6 |
| scTab | 49.8 ± 0.4 | 75.2 ± 0.3 | 56.8 ± 0.5 | 84.1 ± 0.7 | 75.4 | 17.7 | 75.7 | 21.2 |
| scVI [DiverseRNA-1.4M] | 47.0 ± 0.4 | 72.4 ± 0.4 | 51.7 ± 1.1 | 60.4 ± 1.4 | 74.2 | 20.9 | 76.3 | 25.5 |
| STATE | 50.8 ± 0.1 | 78.2 ± 0.2 | 59.2 ± 1.6 | 89.2 ± 0.4 | 80.1 | 38.2 | 81.7 | 40.2 |
| Transcriptformer | 49.7 ± 0.3 | 73.4 ± 0.2 | 51.1 ± 1.4 | 71.2 ± 0.9 | 78.5 | 33.3 | 79.8 | 34.1 |
| TxFM-B [TF-Sapiens data] | 52.6 ± 0.2 | 75.8 ± 0.4 | 53.4 ± 2.1 | 69.4 ± 1.0 | 76.6 | 28.9 | 79.8 | 33.2 |
| TxFM-S [DiverseRNA-1.4M] | 52.1 ± 0.3 | 75.2 ± 0.5 | 55.1 ± 1.1 | 76.2 ± 0.9 | 77.4 | 29.2 | 77.8 | 32.4 |
| TxFM-B [DiverseRNA-1.4M] | 51.6 ± 0.2 | 74.9 ± 0.3 | 55.6 ± 0.7 | 73.5 ± 1.0 | 78.6 | 32.0 | 79.4 | 33.8 |

**Adjusted Rand Index** (ARI): Measures pairwise agreement between clusterings, corrected for chance:

$$\text{ARI} = \frac{\text{RI} - \mathbb{E}[\text{RI}]}{\max(\text{RI}) - \mathbb{E}[\text{RI}]} \tag{15}$$

where $\text{RI} = \frac{n_S + n_D}{\binom{n}{2}}$ counts pairs that are consistently assigned (same cluster in both clusterings: $n_S$; different clusters in both: $n_D$). ARI is more conservative than NMI, requiring precise boundary correspondence for high scores.

**Average Silhouette Width** (ASW)**:** Measures separation of true cell types in embedding space:

$$\text{ASW} = \frac{1}{n} \sum_{i=1}^{n} \frac{b_i^{(l)} - a_i^{(l)}}{\max(a_i^{(l)}, b_i^{(l)})} \tag{16}$$

For each cell $i$ with true label $\ell_i$, $a_i^{(l)}$ is the mean distance to other cells with the same label, and $b_i^{(l)}$ is the mean distance to cells from the nearest different label. High ASW indicates well-separated cell types.

### B.2.2 BATCH INTEGRATION TASK

Batch integration evaluates the dual challenge of removing technical artifacts (batch effects) while preserving biological signal. Batches represent different experimental conditions that should ideally contain identical cell type distributions. Results are obtained by subsampling 10,000 cells with 10 random seeds and reporting the average. Two metrics assess different aspects of this balance:

**Batch-Corrected Biological Preservation** ($\text{ASW}_{l/b}$)**:** Measures whether cell types remain distinguishable within individual batches:

$$\text{ASW}_{l/b} = \frac{1}{n} \sum_{i=1}^{n} \frac{b_i^{(l/b)} - a_i^{(l/b)}}{\max(a_i^{(l/b)}, b_i^{(l/b)})} \tag{17}$$

For each cell $i$ with label $\ell_i$ and batch $\beta_i$, $a_i^{(l/b)}$ is the mean distance to other cells with the same label within the same batch, and $b_i^{(l/b)}$ is the mean distance to cells with different labels but within the same batch. This metric restricts comparisons to within-batch only, testing whether biological structure generalizes across experimental conditions.

**Principal Component Regression comparison** ($\text{PCR}_c$)**:** Measures batch effect reduction by comparing variance explained by batch labels before and after integration [5] :

$$\text{PCR}_c = \max\left(0, \frac{R_{\text{pre}}^2 - R_{\text{post}}^2}{R_{\text{pre}}^2}\right) \tag{18}$$

where $R_{\text{pre}}^2$ and $R_{\text{post}}^2$ are the variance in the first 50 principal components explained by batch labels in the log-normalized data and embedding spaces, respectively. The $\max(0, \cdot)$ operation ensures the score is non-negative when integration increases batch effects. Values closer to 1 indicate higher batch effect removal.

### B.2.3 CLASSIFICATION PROBING TASK

While not part of the original benchmark by Kedzierska et al. (2025), classification probing provides a complementary assessment of representation quality. Linear probes evaluate whether cell type information is linearly separable in the embedding space, while MLP probes test if this information can be recovered through simple non-linear transformations. The probe's performance reflects how well the model has organized biologically meaningful structure within its learned representations.

We follow an evaluation protocol with batch-aware data splitting to prevent information leakage. Data is split at the batch level with 75% for training and 25% for testing, ensuring no batch appears in both sets. Within each training fold, we further reserve 20% of training data for validation using random sampling (not batch-aware).

For preprocessing, we apply StandardScaler fit on training data and applied to test data. The training protocol employs early stopping based on validation loss and ReduceLROnPlateau scheduling. Model selection is performed through hyperparameter grid search over learning rates $\{1 \times 10^{-4}, 5 \times$

---

[5]While the original benchmark implementation used a single call to `pcr()`, which measures absolute remaining batch effects, the paper explicitly describes PCR as comparing "the proportion of the variance that is explained by the batch variable between the original dataset and the embeddings of the model", which would instead correspond to `pcr_comparison()`. With either implementation, PCR values are highly variable and inconsistent between models and datasets. Due to this ambiguity, we opted not to report this metric.

$10^{-4}, 1 \times 10^{-5}$} with fixed weight decay $1 \times 10^{-4}$, using AdamW optimizer and batch size 2048. The configuration maximizing the combined metric $(\text{Acc} + \text{F1 macro})/2$ is selected.

We evaluate two architectures: linear probes and 2-layer MLPs with hidden dimensions $[512, 256]$ and dropout rate $0.5$. Performance is measured using two complementary metrics:

**Accuracy** ($\text{Acc}$)**:**

$$\text{Acc} = \frac{\text{Number of correct predictions}}{\text{Total number of predictions}}$$

**Macro-averaged F1 Score** ($\text{F1 macro}$)**:**

$$\text{F1 macro} = \frac{1}{C} \sum_{c=1}^{C} \text{F1}_c$$

where $\text{F1}_c = \frac{2 \cdot \text{Precision}_c \cdot \text{Recall}_c}{\text{Precision}_c + \text{Recall}_c}$ for class $c$, and $C$ is the number of classes. Unlike accuracy, $\text{F1 macro}$ provides equal weight to all cell types regardless of frequency, making it particularly suitable for imbalanced single-cell datasets where rare cell types are as important as abundant ones.

### B.3 REPRESENTATION LEARNING BENCHMARKS

**Gene-gene relationship recall of non-perturbational gene representations.** In order to obtain the TxFM codebook and decoder relationship recall results presented in § 5.2 and Figure 2, we apply a simple post-processing pipeline. We first standardize each set of gene representations and then fit and transform with PCA, which slightly improves recall over the raw representations.

For comparisons with baselines, we used the scVI model trained on DiverseRNA-1.4M with n_latent=768, n_hidden=1024, AdamW optimizer, lr=0.0005, negative binomial loss, and other settings set to default values (more details in § B.1.2). We extracted the fully connected weight matrix layer of z_encoder as the codebook-equivalent layer and the weight matrix of px_scale_decoder as the the decoder-equivalent layer. For PCA, we use its weight matrix.

In Figure 6 we observe that for TxFM-B Default, the codebook representations surprisingly achieve the highest recall when PCA is set to only 32 components, while the decoder representations have their best recall at 256 components; this trend was consistent across different TxFM versions trained with different hyperparameters. Interestingly, this is in contrast to scVI, which consistently improved as the number of PCA components increased. As such, for each recall value reported for TxFM codebook and decoder benchmarks, we apply PCA to 32 components and to 256 components respectively (while keeping to 1024 components for scVI and the PCA baseline), and then compute relationship recall on those post-processed lower-dimensional representations. The recall is calculated 3 times against different random null distributions in order to obtain a mean and standard deviation of performance for testing statistical significance.

## C ABLATION EXPERIMENTS

When testing differences in perturbation consistency (perts), recall for encoder gene tokens (enc), and recall for decoder gene weights (dec) between different versions of TxFM, we compared the metric corresponding to each ablation against the Default setting. For perts comparisons, we used a 2-proportion z-test. For enc and dec comparisons, we used a 2-tailed t-test. All resulting p-values were adjusted with Benjamini-Hochberg multiple test correction at 0.05 error rate across all pairs of ablations within each of the three metrics.

### C.1 ACTIVATION FUNCTIONS

We use a novel activation function called *rectified tanh* (Equation 1) in the decoder to safeguard against predicting counts outside of the training distribution, by leveraging the normalization procedure we employ in preprocessing. Our activation function asymptotically tends to the library size, i.e. the largest possible value an entire expression vector of a preprocessed sample can sum up to. We add 1 to the library size $L$ to mirror the pseudocounts added to input data in our preprocessing. (The activation can be generalized to $\phi(z) = \alpha \text{ReLU}(\tanh(\frac{z}{\beta}))$ and unit-size gradient steps can be

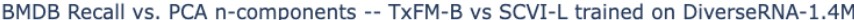
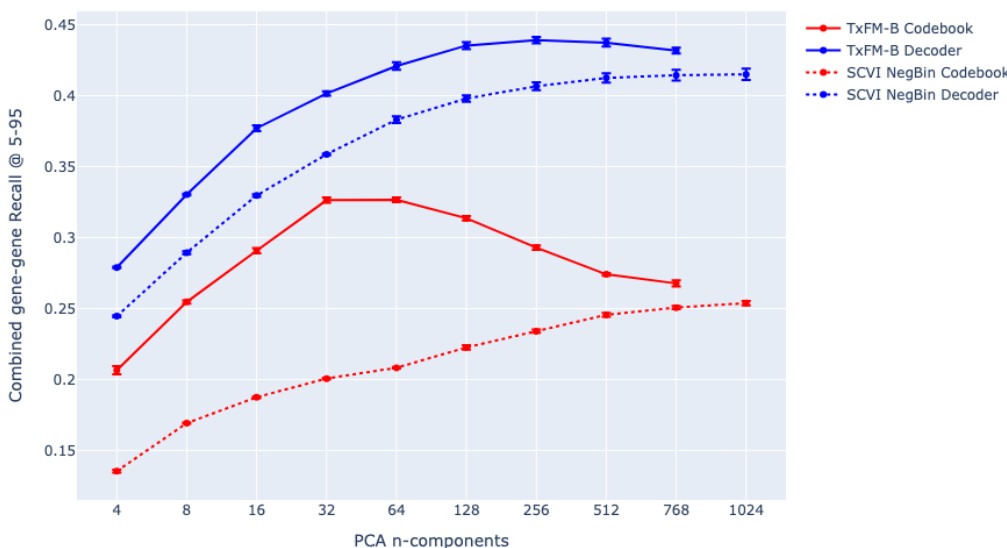

Figure 6: **TxFM encoder and decoder representations have different intrinsic ranks for optimal biological relationship recall.** Gene-gene relationship recall computed using a 5-95% null distribution percentile thresholds (Kraus et al., 2024)) over 5 relationship databases combined spanning the whole genome (~17,000 genes). We evaluate the PCA-transformed gene representations extracted from TxFM-B codebook and decoder and the equivalent layers of scVI, and measure recall as a function of the number of principal components. Note that these are representations of *genes as features* and not as perturbations.

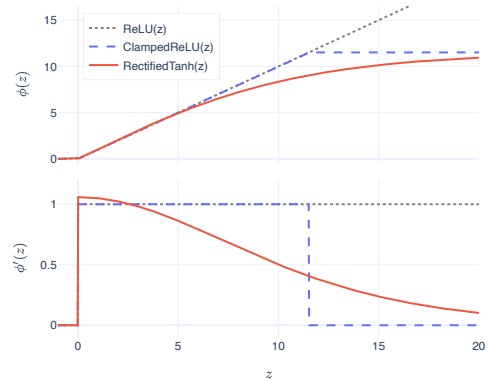
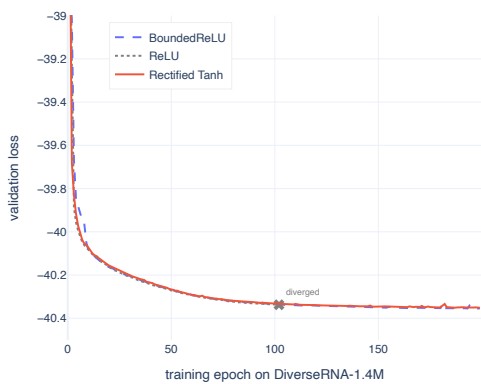

(a) Count activation functions and their derivatives on logits $z$.

(b) Model validation loss per epoch.

Figure 7: Comparing TxFM decoder count activation functions for models trained in ablation (f). A pure ReLU activation diverges to NaN midway through training. Clamping ReLU prevents divergence and achieves comparable loss, but has lower zero-shot performance in representing perturbations compared to our rectified tanh activation function (Table 5).

ensured by setting $\alpha = \beta$; we recommend setting $\alpha = \log(L + 1)$ where $L$ is the library size the data is pre-normalized to.)

Indeed, given that the count data is always non-negative, we first tried ReLU as the count activation function $\phi$, but, halfway through training, the model diverges. While a softmax activation struggled to train effectively, we found that simply clamping the ReLU activation based on the maximum possible library size could prevent divergence:

$$\phi_{ClampedReLU}(z) = \min(\text{ReLU}(z), \log(L + 1)).$$

Figure 7 compares rectified tanh to a ReLU and clamped ReLU. Our novel activation function achieved the best zero-shot performance in perturbation representation (Table 5).

## C.2 DATASET CURATION

We evaluate these dataset curation strategies in Table 5 (h):

1. Removing all K562 cells from the training data, yielding 932K training samples.

2. Adding only the 72,000 non-targeting control K562 cells from the Replogle et al. (2022) dataset, in addition to the 932K non-K562 samples in the DiverseRNA dataset.

3. Our default approach, which follows an established strategy shown to improve the performance of MAEs on biological experimental data (Kenyon-Dean et al., 2025) by filtering the perturbational K562 training data to only include cells from perturbations distinguishable from the rest. Specifically, we selected *phenoprint* perturbations by using an earlier version of TxFM to inference the data, computed perturbation consistency as described above (§ B.3), and kept perturbations with $p < 0.1$. This more permissive threshold allows us to increase the number of training samples while still keeping perturbations with reasonably distinct transcriptional profiles.

4. Training with the full uncurated dataset of K562 cells, yielding 2.8M training samples, and holding the compute budget constant by adjusting the number of training epochs according to dataset size; we also evaluate doubling the training compute on the 2.8M dataset to 200 epochs [2x], to about 2,000 H100 GPU hours.

5. Training TxFM-B with 4x more compute on a different dataset of 57 million cells sampled from 72 large-scale atlases from CZI et al. (2025).

Table 8: **Backbone choice for existing models**. Average score across all benchmarking tasks in Bendidi et al. (2024) on RPE1, HEPG2, and Jurkat datasets for different backbone choices of existing FMs with multiple pretrained backbones. Grayed backbones are the best performing backbones used in Table 2.

| Backbone | RPE1 | HEPG2 | Jurkat |
|---|---|---|---|
| GenePT ADA | 39.29 | 35.45 | 32.07 |
| GenePT Large | 39.82 | 34.77 | 32.55 |

(a) **GenePT** ablation of best performing backbone.

| Backbone | RPE1 | HEPG2 | Jurkat |
|---|---|---|---|
| AIDO.cell 3M | 28.03 | 29.21 | 25.87 |
| AIDO.cell 10M | 31.01 | 31.84 | 26.47 |
| AIDO.cell 100M | 37.45 | 34.52 | 31.53 |

(b) **AIDO.cell** ablation of best performing backbone.

| Backbone | RPE1 | HEPG2 | Jurkat |
|---|---|---|---|
| scPrint Small | 25.70 | 22.03 | 24.15 |
| scPrint Medium | 30.84 | 24.22 | 26.58 |
| scPrint Large | 31.62 | 23.80 | 27.04 |

(c) **scPrint** ablation of best performing backbone.

| Backbone | RPE1 | HEPG2 | Jurkat |
|---|---|---|---|
| Transcriptformer Exemplar | 34.53 | 31.54 | 29.96 |
| Transcriptformer Metazoa | 34.52 | 31.51 | 30.00 |
| Transcriptformer Sapiens | 34.88 | 31.55 | 29.92 |

(d) **TranscriptFormer** ablation of best performing backbone.

# D    DATASET-SPECIFIC BENCHMARKING RESULTS

Figure 8 and Table 8 show ablations of the best backbone for each existing model and the best performing number of principal components for PCA on this benchmark. Tables 9, 10, 11 show the results on the Bendidi et al. (2024) benchmark for the RPE-1, HEPG2, and Jurkat datasets, correspondingly.

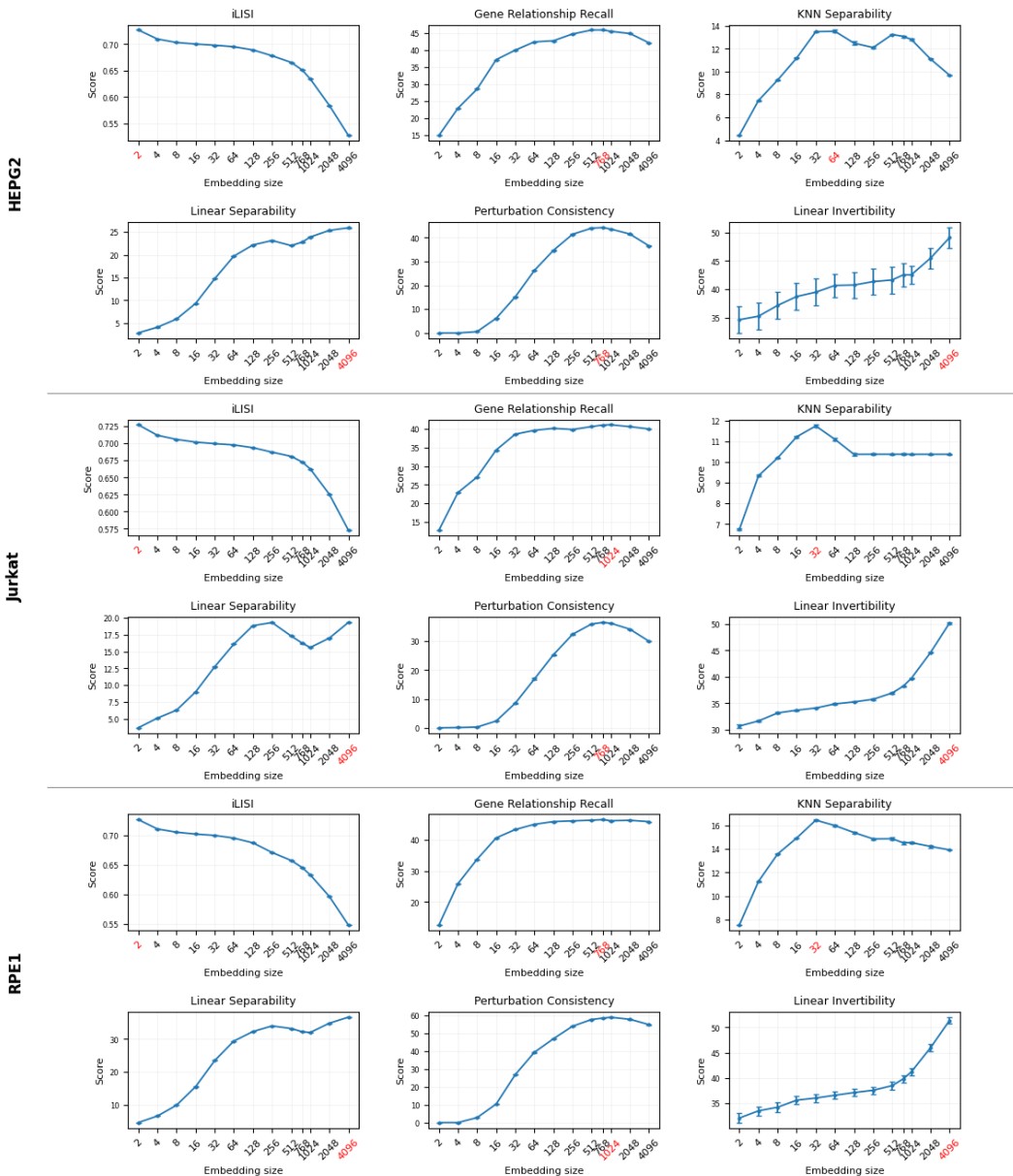

Figure 8: **Dimension size ablation for PCA.** Bendidi et al. (2024) benchmarking results on RPE1, HEPG2, and Jurkat datasets after applying PCA on raw counts with different numbers of principal components. Red is optimal size for each downstream task.

Table 9: Overview of model performance across different settings and perturbational downstream tasks on the scores for **RPE1** dataset. Metrics are mean $\pm$ std across 5 random seeds.

| Models | ilisi↑ | lin↑ | knn↑ | p.cst↑ | bmdb↑ | inv↑ | avg↑ |
|---|---|---|---|---|---|---|---|
| **Fit on evaluation data** (unsupervised) | | | | | | | |
| PCA | $0.63_{\pm0.00}$ | $32.03_{\pm0.04}$ | $14.54_{\pm0.05}$ | $58.79_{\pm0.1}$ | $46.03_{\pm0.04}$ | $41.17_{\pm0.7}$ | 42.64 |
| scVI | $0.69_{\pm0.00}$ | $19.68_{\pm0.01}$ | $12.69_{\pm0.03}$ | $38.27_{\pm0.1}$ | $41.86_{\pm0.04}$ | $43.93_{\pm0.7}$ | 37.65 |
| TxFM-B finetuned (ours) | $0.70_{\pm0.00}$ | $42.10_{\pm0.01}$ | $17.82_{\pm0.04}$ | $48.73_{\pm0.2}$ | $47.27_{\pm0.04}$ | $42.64_{\pm0.7}$ | 44.85 |
| **Count data baselines** | | | | | | | |
| Random label shuffle | $0.70_{\pm0.00}$ | $0.85_{\pm0.01}$ | $8.20_{\pm0.01}$ | $0.00_{\pm0.00}$ | $11.17_{\pm0.02}$ | $25.34_{\pm0.8}$ | 19.40 |
| Raw data | $0.61_{\pm0.00}$ | $36.68_{\pm0.01}$ | $13.92_{\pm0.03}$ | $36.23_{\pm0.1}$ | $44.06_{\pm0.05}$ | $59.63_{\pm0.5}$ | 41.93 |
| (Lib+Log)Norm | $0.61_{\pm0.00}$ | $30.07_{\pm0.01}$ | $14.74_{\pm0.03}$ | $37.60_{\pm0.1}$ | $42.37_{\pm0.06}$ | $62.18_{\pm0.6}$ | 41.37 |
| (Lib+Log)Norm+5k HVG | $0.64_{\pm0.00}$ | $26.66_{\pm0.01}$ | $14.17_{\pm0.04}$ | $41.61_{\pm0.2}$ | $43.24_{\pm0.06}$ | $52.02_{\pm0.6}$ | 40.41 |
| (Lib+Log)Norm+1024HVG | $0.69_{\pm0.00}$ | $23.67_{\pm0.01}$ | $14.02_{\pm0.03}$ | $36.73_{\pm0.1}$ | $43.45_{\pm0.06}$ | $41.25_{\pm0.6}$ | 38.06 |
| **ChatGPT Embeddings** | | | | | | | |
| GenePT-Large | $0.73_{\pm0.00}$ | $27.49_{\pm0.01}$ | $12.14_{\pm0.04}$ | $43.46_{\pm0.2}$ | $44.35_{\pm0.08}$ | $37.55_{\pm0.7}$ | 39.82 |
| **Pretrained FMs** | | | | | | | |
| Geneformer [CELLxGENE] | $0.71_{\pm0.00}$ | $4.08_{\pm0.01}$ | $8.22_{\pm0.02}$ | $0.00_{\pm0.00}$ | $11.04_{\pm0.02}$ | $25.80_{\pm0.7}$ | 20.15 |
| GeneJEPA [Tahoe-100M] | $0.70_{\pm0.00}$ | $4.68_{\pm0.00}$ | $8.37_{\pm0.03}$ | $0.45_{\pm0.00}$ | $22.19_{\pm0.03}$ | $36.17_{\pm0.2}$ | 23.74 |
| scTab [CELLxGENE] | $0.70_{\pm0.00}$ | $6.69_{\pm0.01}$ | $8.76_{\pm0.03}$ | $1.73_{\pm0.03}$ | $31.60_{\pm0.05}$ | $35.33_{\pm0.7}$ | 25.80 |
| CellPLM | $0.71_{\pm0.00}$ | $7.32_{\pm0.00}$ | $9.50_{\pm0.06}$ | $2.18_{\pm0.01}$ | $23.74_{\pm0.03}$ | $36.89_{\pm0.7}$ | 25.20 |
| UCE [CELLxGENE] | $0.70_{\pm0.00}$ | $7.42_{\pm0.00}$ | $9.34_{\pm0.05}$ | $10.24_{\pm0.1}$ | $29.36_{\pm0.03}$ | $36.74_{\pm0.7}$ | 27.34 |
| scCello [CELLxGENE] | $0.71_{\pm0.00}$ | $14.04_{\pm0.01}$ | $10.52_{\pm0.1}$ | $6.37_{\pm0.0}$ | $31.03_{\pm0.03}$ | $37.16_{\pm0.8}$ | 27.53 |
| scPrint-L [CELLxGENE] | $0.70_{\pm0.00}$ | $16.26_{\pm0.01}$ | $11.38_{\pm0.04}$ | $16.38_{\pm0.1}$ | $37.59_{\pm0.03}$ | $37.44_{\pm0.7}$ | 31.62 |
| scGPT [CELLxGENE] | $0.71_{\pm0.00}$ | $12.58_{\pm0.01}$ | $10.34_{\pm0.04}$ | $18.26_{\pm0.1}$ | $34.19_{\pm0.03}$ | $37.55_{\pm0.7}$ | 30.67 |
| Tahoe-x1 [Tahoe-100M] | $0.71_{\pm0.00}$ | $17.16_{\pm0.01}$ | $11.59_{\pm0.02}$ | $22.64_{\pm0.3}$ | $37.23_{\pm0.03}$ | $39.16_{\pm0.2}$ | 33.15 |
| TranscriptFormer-Sapiens [CELLxGENE] | $0.71_{\pm0.00}$ | $18.65_{\pm0.01}$ | $10.90_{\pm0.03}$ | $34.22_{\pm0.1}$ | $35.08_{\pm0.03}$ | $39.40_{\pm0.7}$ | 34.88 |
| AIDO.Cell-100M [CELLxGENE] | $0.71_{\pm0.00}$ | $27.66_{\pm0.01}$ | $12.88_{\pm0.03}$ | $31.62_{\pm0.04}$ | $42.61_{\pm0.03}$ | $38.54_{\pm0.7}$ | 37.45 |
| Cell2Sentence | $0.70_{\pm0.00}$ | $26.73_{\pm0.00}$ | $13.10_{\pm0.04}$ | $40.09_{\pm0.1}$ | $43.38_{\pm0.03}$ | $39.83_{\pm0.2}$ | 39.01 |
| TxFM-B [TF-Sapiens data] (ours) | $0.71_{\pm0.00}$ | $27.54_{\pm0.01}$ | $12.55_{\pm0.03}$ | $36.48_{\pm0.1}$ | $42.11_{\pm0.04}$ | $41.47_{\pm0.7}$ | 38.57 |
| STATE-SE [CxG + Tahoe-100M + scBC] | $0.69_{\pm0.00}$ | $29.05_{\pm0.01}$ | $13.31_{\pm0.1}$ | $40.80_{\pm0.2}$ | $44.59_{\pm0.04}$ | $44.97_{\pm0.2}$ | 40.38 |
| **Fit on our public train set** | | | | | | | |
| PCA [DiverseRNA-1.4M] | $0.69_{\pm0.00}$ | $16.10_{\pm0.01}$ | $12.65_{\pm0.02}$ | $29.59_{\pm0.2}$ | $40.01_{\pm0.04}$ | $46.38_{\pm0.6}$ | 35.72 |
| scVI [DiverseRNA-1.4M] | $0.71_{\pm0.00}$ | $22.10_{\pm0.01}$ | $13.08_{\pm0.03}$ | $36.25_{\pm0.1}$ | $39.07_{\pm0.03}$ | $39.52_{\pm0.7}$ | 35.18 |
| TxFM-S [DiverseRNA-1.4M] (ours) | $0.71_{\pm0.00}$ | $35.11_{\pm0.01}$ | $16.04_{\pm0.03}$ | $42.10_{\pm0.4}$ | $46.21_{\pm0.04}$ | $41.06_{\pm0.3}$ | 41.94 |
| TxFM-B [DiverseRNA-1.4M] (ours) | $0.70_{\pm0.00}$ | $37.61_{\pm0.01}$ | $15.13_{\pm0.03}$ | $42.71_{\pm0.6}$ | $45.33_{\pm0.04}$ | $41.54_{\pm0.7}$ | 42.17 |

Table 10: Overview of model performance across different settings and perturbational downstream tasks on the scores for **HEPG2** dataset. Metrics are mean $\pm$ std across 5 random seeds.

| Models | ilisi↑ | lin↑ | knn↑ | p.cst↑ | bmdb↑ | inv↑ | avg↑ |
|---|---|---|---|---|---|---|---|
| **Fit on evaluation data** (unsupervised) | | | | | | | |
| PCA | $0.63_{\pm 0.00}$ | $23.90_{\pm 0.00}$ | $12.81_{\pm 0.04}$ | $43.56_{\pm 0.14}$ | $45.57_{\pm 0.04}$ | $42.54_{\pm 1.53}$ | 38.63 |
| scVI | $0.69_{\pm 0.00}$ | $12.61_{\pm 0.01}$ | $11.47_{\pm 0.12}$ | $27.84_{\pm 0.05}$ | $45.22_{\pm 0.04}$ | $47.03_{\pm 2.23}$ | 35.63 |
| TxFM-B finetuned (ours) | $0.70_{\pm 0.00}$ | $29.27_{\pm 0.01}$ | $15.81_{\pm 0.03}$ | $38.40_{\pm 0.14}$ | $46.24_{\pm 0.04}$ | $44.55_{\pm 3.14}$ | 40.78 |
| **Count data baselines** | | | | | | | |
| Random label shuffle | $0.70_{\pm 0.00}$ | $0.66_{\pm 0.01}$ | $5.45_{\pm 0.05}$ | $0.00_{\pm 0.00}$ | $12.95_{\pm 0.02}$ | $28.14_{\pm 2.51}$ | 19.68 |
| Raw data | $0.60_{\pm 0.00}$ | $23.21_{\pm 0.79}$ | $9.75_{\pm 0.06}$ | $2.60_{\pm 0.12}$ | $32.15_{\pm 0.05}$ | $56.38_{\pm 2.03}$ | 30.81 |
| (Lib+Log)Norm | $0.65_{\pm 0.00}$ | $21.51_{\pm 0.01}$ | $11.93_{\pm 0.04}$ | $10.49_{\pm 0.23}$ | $35.32_{\pm 0.06}$ | $60.09_{\pm 1.63}$ | 34.12 |
| (Lib+Log)Norm+5k HVG | $0.67_{\pm 0.00}$ | $19.22_{\pm 0.01}$ | $11.38_{\pm 0.08}$ | $8.09_{\pm 0.13}$ | $34.75_{\pm 0.06}$ | $50.54_{\pm 2.03}$ | 31.88 |
| (Lib+Log)Norm+1024HVG | $0.69_{\pm 0.00}$ | $15.04_{\pm 0.01}$ | $10.74_{\pm 0.04}$ | $24.13_{\pm 0.04}$ | $44.06_{\pm 0.04}$ | $44.46_{\pm 2.13}$ | 34.65 |
| **ChatGPT Embeddings** | | | | | | | |
| GenePT-Large | $0.73_{\pm 0.00}$ | $18.65_{\pm 0.08}$ | $8.28_{\pm 0.03}$ | $24.78_{\pm 0.04}$ | $42.01_{\pm 0.04}$ | $40.91_{\pm 2.46}$ | 34.77 |
| **Pretrained FMs** | | | | | | | |
| Geneformer [CELLxGENE] | $0.71_{\pm 0.00}$ | $2.21_{\pm 0.005}$ | $5.44_{\pm 0.02}$ | $0.04_{\pm 0.00}$ | $11.30_{\pm 0.02}$ | $28.41_{\pm 2.44}$ | 19.89 |
| GeneJEPA [Tahoe-100M] | $0.70_{\pm 0.00}$ | $2.64_{\pm 0.005}$ | $5.74_{\pm 0.00}$ | $0.41_{\pm 0.03}$ | $23.68_{\pm 0.03}$ | $38.86_{\pm 1.82}$ | 23.64 |
| scTab [CELLxGENE] | $0.70_{\pm 0.00}$ | $4.59_{\pm 0.026}$ | $6.67_{\pm 0.02}$ | $1.93_{\pm 0.03}$ | $32.89_{\pm 0.03}$ | $38.80_{\pm 2.32}$ | 25.91 |
| CellPLM | $0.71_{\pm 0.005}$ | $5.12_{\pm 0.027}$ | $7.52_{\pm 0.03}$ | $3.68_{\pm 0.07}$ | $23.03_{\pm 0.04}$ | $40.68_{\pm 2.22}$ | 25.25 |
| UCE [CELLxGENE] | $0.70_{\pm 0.005}$ | $5.23_{\pm 0.017}$ | $7.20_{\pm 0.03}$ | $6.31_{\pm 0.01}$ | $31.32_{\pm 0.04}$ | $40.47_{\pm 2.32}$ | 26.92 |
| scCello [CELLxGENE] | $0.71_{\pm 0.008}$ | $8.50_{\pm 0.037}$ | $7.44_{\pm 0.03}$ | $6.95_{\pm 0.03}$ | $36.77_{\pm 0.04}$ | $40.14_{\pm 2.33}$ | 28.50 |
| scPrint-L [CELLxGENE] | $0.71_{\pm 0.002}$ | $2.96_{\pm 0.005}$ | $5.93_{\pm 0.03}$ | $0.16_{\pm 0.00}$ | $26.89_{\pm 0.03}$ | $35.57_{\pm 2.42}$ | 23.80 |
| scGPT [CELLxGENE] | $0.70_{\pm 0.007}$ | $7.21_{\pm 0.018}$ | $8.51_{\pm 0.04}$ | $11.46_{\pm 0.03}$ | $36.69_{\pm 0.04}$ | $41.30_{\pm 2.22}$ | 29.35 |
| Tahoe-x1 [Tahoe-100M] | $0.70_{\pm 0.009}$ | $9.53_{\pm 0.029}$ | $9.83_{\pm 0.05}$ | $15.39_{\pm 0.03}$ | $39.30_{\pm 3.24}$ | $43.25_{\pm 1.53}$ | 31.36 |
| TranscriptFormer-Sapiens [CELLxGENE] | $0.70_{\pm 0.00}$ | $12.16_{\pm 0.08}$ | $8.94_{\pm 0.04}$ | $19.33_{\pm 0.03}$ | $35.14_{\pm 0.04}$ | $42.77_{\pm 2.22}$ | 31.55 |
| AIDO.Cell-100M [CELLxGENE] | $0.71_{\pm 0.00}$ | $16.98_{\pm 0.01}$ | $10.26_{\pm 0.02}$ | $25.05_{\pm 0.04}$ | $41.42_{\pm 0.04}$ | $42.28_{\pm 2.22}$ | 34.52 |
| Cell2Sentence | $0.70_{\pm 0.00}$ | $16.12_{\pm 0.09}$ | $9.99_{\pm 0.08}$ | $24.52_{\pm 0.04}$ | $42.24_{\pm 0.04}$ | $43.44_{\pm 1.83}$ | 34.49 |
| TxFM-B [TF-Sapiens data] (ours) | $0.70_{\pm 0.00}$ | $17.81_{\pm 0.01}$ | $10.59_{\pm 0.02}$ | $25.28_{\pm 0.14}$ | $43.31_{\pm 0.04}$ | $43.87_{\pm 2.44}$ | 35.30 |
| STATE-SE [CxG + Tahoe-100M + scBC] | $0.69_{\pm 0.00}$ | $17.20_{\pm 0.00}$ | $10.79_{\pm 0.02}$ | $23.13_{\pm 0.04}$ | $43.94_{\pm 0.04}$ | $45.15_{\pm 1.73}$ | 35.10 |
| **Fit on our public train set** | | | | | | | |
| PCA [DiverseRNA-1.4M] | $0.69_{\pm 0.00}$ | $11.11_{\pm 0.01}$ | $11.50_{\pm 0.02}$ | $20.53_{\pm 0.14}$ | $43.34_{\pm 0.04}$ | $48.21_{\pm 2.13}$ | 34.00 |
| scVI [DiverseRNA-1.4M] | $0.70_{\pm 0.00}$ | $14.16_{\pm 0.01}$ | $10.83_{\pm 0.00}$ | $17.28_{\pm 0.04}$ | $40.00_{\pm 0.04}$ | $43.17_{\pm 2.22}$ | 32.69 |
| TxFM-S [DiverseRNA-1.4M] (ours) | $0.70_{\pm 0.00}$ | $23.08_{\pm 0.01}$ | $13.11_{\pm 0.03}$ | $29.52_{\pm 0.04}$ | $45.88_{\pm 0.04}$ | $44.56_{\pm 1.83}$ | 37.82 |
| TxFM-B [DiverseRNA-1.4M] (ours) | $0.70_{\pm 0.00}$ | $25.93_{\pm 0.01}$ | $13.14_{\pm 0.03}$ | $32.47_{\pm 0.24}$ | $45.26_{\pm 0.04}$ | $44.40_{\pm 2.23}$ | 38.63 |

Table 11: Overview of model performance across different settings and perturbational downstream tasks on the scores for **Jurkat** dataset. Metrics are mean $\pm$ std across 5 random seeds.

| Models | ilisi↑ | lin↑ | knn↑ | p.cst↑ | bmdb↑ | inv↑ | avg↑ |
|---|---|---|---|---|---|---|---|
| **Fit on evaluation data** (unsupervised) | | | | | | | |
| PCA | $0.66_{\pm0.00}$ | $15.59_{\pm0.01}$ | $10.37_{\pm0.03}$ | $36.27_{\pm0.14}$ | $41.13_{\pm0.03}$ | $39.70_{\pm0.13}$ | 34.89 |
| scVI | $0.69_{\pm0.00}$ | $9.95_{\pm0.01}$ | $9.92_{\pm0.08}$ | $23.26_{\pm0.1}$ | $40.50_{\pm0.04}$ | $42.39_{\pm0.03}$ | 32.62 |
| TxFM-B finetuned (ours) | $0.69_{\pm0.00}$ | $25.64_{\pm0.01}$ | $11.99_{\pm0.03}$ | $36.56_{\pm0.1}$ | $42.64_{\pm0.04}$ | $41.52_{\pm0.1}$ | 38.01 |
| **Count data baselines** | | | | | | | |
| Random label shuffle | $0.70_{\pm0.00}$ | $0.54_{\pm0.00}$ | $7.46_{\pm0.04}$ | $0.04_{\pm0.00}$ | $11.59_{\pm0.02}$ | $24.77_{\pm0.01}$ | 19.20 |
| Raw data | $0.63_{\pm0.00}$ | $20.47_{\pm0.05}$ | $10.36_{\pm0.01}$ | $11.55_{\pm0.03}$ | $35.25_{\pm0.05}$ | $57.28_{\pm0.13}$ | 33.06 |
| (Lib+Log)Norm | $0.62_{\pm0.00}$ | $16.73_{\pm0.01}$ | $11.07_{\pm0.01}$ | $14.69_{\pm0.03}$ | $34.71_{\pm0.06}$ | $60.50_{\pm0.13}$ | 33.37 |
| (Lib+Log)Norm+5k HVG | $0.64_{\pm0.00}$ | $13.07_{\pm0.01}$ | $10.76_{\pm0.02}$ | $20.95_{\pm0.02}$ | $37.29_{\pm0.07}$ | $50.61_{\pm0.1}$ | 32.94 |
| (Lib+Log)Norm+1024HVG | $0.68_{\pm0.00}$ | $9.67_{\pm0.01}$ | $10.38_{\pm0.01}$ | $16.95_{\pm0.1}$ | $38.68_{\pm0.01}$ | $39.44_{\pm0.03}$ | 30.67 |
| **ChatGPT Embeddings** | | | | | | | |
| GenePT-Large | $0.74_{\pm0.00}$ | $14.30_{\pm0.08}$ | $8.58_{\pm0.01}$ | $24.47_{\pm0.1}$ | $38.11_{\pm0.03}$ | $35.74_{\pm0.43}$ | 32.55 |
| **Pretrained FMs** | | | | | | | |
| Geneformer [CELLxGENE] | $0.71_{\pm0.00}$ | $2.97_{\pm0.00}$ | $7.44_{\pm0.03}$ | $0.00_{\pm0.00}$ | $10.60_{\pm0.02}$ | $25.00_{\pm0.31}$ | 19.63 |
| GeneJEPA [Tahoe-100M] | $0.70_{\pm0.003}$ | $3.14_{\pm0.01}$ | $7.59_{\pm0.04}$ | $0.04_{\pm0.00}$ | $14.96_{\pm0.03}$ | $33.56_{\pm0.22}$ | 21.66 |
| scTab [CELLxGENE] | $0.70_{\pm0.003}$ | $3.70_{\pm0.01}$ | $7.72_{\pm0.04}$ | $0.12_{\pm0.00}$ | $21.12_{\pm0.02}$ | $32.79_{\pm0.12}$ | 22.69 |
| CellPLM | $0.71_{\pm0.004}$ | $4.27_{\pm0.00}$ | $8.19_{\pm0.03}$ | $1.79_{\pm0.07}$ | $23.99_{\pm0.03}$ | $34.81_{\pm0.1}$ | 24.06 |
| UCE [CELLxGENE] | $0.71_{\pm0.004}$ | $4.97_{\pm0.00}$ | $7.97_{\pm0.02}$ | $2.95_{\pm0.01}$ | $28.61_{\pm0.03}$ | $34.87_{\pm0.12}$ | 25.07 |
| scCello [CELLxGENE] | $0.70_{\pm0.00}$ | $7.26_{\pm0.01}$ | $8.69_{\pm0.04}$ | $0.37_{\pm0.00}$ | $28.45_{\pm0.03}$ | $34.77_{\pm0.02}$ | 25.09 |
| scPrint-L [CELLxGENE] | $0.70_{\pm0.00}$ | $8.43_{\pm0.01}$ | $9.24_{\pm.01}$ | $5.13_{\pm0.06}$ | $33.34_{\pm0.03}$ | $35.48_{\pm0.12}$ | 27.04 |
| scGPT [CELLxGENE] | $0.71_{\pm0.00}$ | $7.66_{\pm0.01}$ | $8.61_{\pm0.05}$ | $5.53_{\pm0.17}$ | $32.45_{\pm0.03}$ | $35.70_{\pm0.12}$ | 26.86 |
| Tahoe-x1 [Tahoe-100M] | $0.71_{\pm0.00}$ | $9.61_{\pm0.01}$ | $9.40_{\pm0.00}$ | $10.15_{\pm0.03}$ | $34.02_{\pm0.03}$ | $36.37_{\pm0.13}$ | 28.47 |
| TranscriptFormer-Sapiens [CELLxGENE] | $0.70_{\pm0.00}$ | $10.77_{\pm0.0}$ | $8.68_{\pm0.04}$ | $20.40_{\pm0.0}$ | $31.38_{\pm0.03}$ | $37.30_{\pm0.12}$ | 29.92 |
| AIDO.Cell-100M [CELLxGENE] | $0.71_{\pm0.00}$ | $15.07_{\pm0.0}$ | $9.46_{\pm.05}$ | $19.54_{\pm0.03}$ | $37.08_{\pm0.03}$ | $36.69_{\pm0.13}$ | 31.53 |
| Cell2Sentence | $0.71_{\pm0.00}$ | $13.93_{\pm0.0}$ | $9.39_{\pm0.04}$ | $26.21_{\pm0.03}$ | $39.11_{\pm0.03}$ | $37.21_{\pm0.03}$ | 32.82 |
| TxFM-B [TF-Sapiens data] (ours) | $0.71_{\pm0.00}$ | $16.52_{\pm0.0}$ | $9.82_{\pm0.05}$ | $22.59_{\pm0.1}$ | $37.97_{\pm0.03}$ | $39.77_{\pm0.13}$ | 32.98 |
| STATE-SE [CxG + Tahoe-100M + scBC] | $0.69_{\pm0.00}$ | $17.15_{\pm0.0}$ | $9.92_{\pm0.04}$ | $24.45_{\pm0.1}$ | $39.44_{\pm0.04}$ | $42.65_{\pm0.03}$ | 33.81 |
| **Fit on our public train set** | | | | | | | |
| PCA [DiverseRNA-1.4M] | $0.69_{\pm0.00}$ | $7.60_{\pm0.00}$ | $10.04_{\pm0.01}$ | $16.83_{\pm0.03}$ | $37.44_{\pm0.04}$ | $44.56_{\pm0.03}$ | 30.91 |
| scVI [DiverseRNA-1.4M] | $0.71_{\pm0.00}$ | $12.97_{\pm0.0}$ | $10.43_{\pm0.01}$ | $16.18_{\pm0.03}$ | $36.16_{\pm0.03}$ | $37.36_{\pm0.1}$ | 30.68 |
| TxFM-S [DiverseRNA-1.4M] (ours) | $0.70_{\pm0.00}$ | $22.18_{\pm0.0}$ | $11.60_{\pm0.02}$ | $28.51_{\pm0.1}$ | $42.56_{\pm0.03}$ | $38.89_{\pm0.03}$ | 35.75 |
| TxFM-B [DiverseRNA-1.4M] (ours) | $0.69_{\pm0.00}$ | $23.79_{\pm0.01}$ | $11.59_{\pm0.03}$ | $31.27_{\pm0.1}$ | $42.08_{\pm0.04}$ | $40.42_{\pm0.13}$ | 36.52 |

