# OpenReview forum: "Effective Biological Representation Learning by Masking Gene Expression"
_ICLR.cc/2026/Workshop/FM4Science — ICLR 2026 Workshop FM4Science Poster_

### Official Review · Reviewer_zDJY · 2026-02-16
**Manuscript Review: Effective Biological Representation Learning by Masking Gene Expression**

**Rating:** 9
**Confidence:** 4

**Review:**

Transcriptomic foundation models (FMs) often struggle to outperform simple baselines despite being trained on massive datasets. To address this, the authors introduce TxFM, a self-supervised transformer masked autoencoder specifically designed for the high-dimensional, unordered nature of gene expression data. Rather than relying on sheer data volume, the authors curated a specialized dataset of 1.4 million bulk and single-cell RNA-seq samples, dubbed DiverseRNA-1.4M. TxFM pre-processes this data using library size normalization and a log-transform. During training, it uniformly masks approximately 90% of the genes. An encoder processes the unmasked genes alongside a CLS token, after which a lightweight MLP decoder attempts to reconstruct the full expression profile solely from the CLS token. The model utilizes a novel rectified tanh activation function and a Poisson-based reconstruction loss to naturally constrain predictions to the bounds of the count-based data.

The authors tackle a recognized issue —that massive transcriptomic FMs often fail to beat simple baselines like PCA. By proving that thoughtful data curation and domain-specific architectural adjustments (like Poisson loss and rectified tanh) matter more than simply scaling up parameters and dataset size, they provide an immensely practical framework for the computational biology community. The benchmarking is transparent, thorough, and clearly supports their claims.

The paper demonstrates that training on a carefully curated 1.4 million sample dataset often outperforms training on uncurated atlases that are 10 to 100 times larger. The authors benchmark TxFM against 16 existing FMs (including scGPT, Geneformer, and scTab) and robust count-based baselines. TxFM establishes a new state-of-the-art for zero-shot genetic perturbation representation across three held-out cell lines. The manuscript includes extensive ablations that effectively isolate the impact of specific architectural choices, such as the 90% mask ratio, decoder depth, and data curation strategies. The model learns meaningful biology without direct supervision, as the learned encoder token embeddings and decoder reconstruction weights successfully recover known gene-gene relationships and protein complexes.

While the base model (TxFM-B) performed exceptionally well, scaling the architecture to a "Large" variant (TxFM-L) on the 1.4M dataset improved validation loss but failed to translate into downstream transfer learning performance gains. Besides, the domain-specific curation may introduce distributional biases that could limit the model's generalization to different biological contexts.

---

### Official Review · Reviewer_wiir · 2026-02-17
**Applied MAE representation learning used in human transcriptome data**

**Rating:** 6
**Confidence:** 3

**Review:**

The authors proposed TxFM, which is an MAE-based foundational model for transcriptomics with the motivation to address the persisting issues in this fields such as lack of generality in unseen cellular settings and failure in zero-shot capability. Although MAE and the parts applied in the framework are not novel, such technical ideas were first time being applied in transcriptomics and showed promising results. Besides the technical foundation of this framework, the authors spent efforts in data curation showing that quality of data set is more important compared with the quantity of data. TxFM achieves state-of-the-art perturbation representation and recovered known gene–gene relationships from its learned parameters. Overall this paper is written with technical foundation and potential publishable insights for future direction.

Major comments:
1. The title is “effective biological representation learning”, however overall the data used in this paper is about human and oncology perturbation representation learning. Such wording need to be refined to be more specific.
2. Reconstruction objective used in MAE has bias. Gene expression is usually zero-inflated. How reconstruction loss address this unique trait in  gene expression?
3. Negative binomial was mentioned but not clear.
4. Scaling from Base (159M) to Large (403M) failed to improve performance, what is the reason for that?.


Minor comments:
1. The authors mainly focus on sc-sequence RNA data, how about bulk-RNA-seq (only small ratio of DiverseRNA-1.4M focus on bulk RNA-seq)? Could this framework have similar outcomes once applied in pure bulk-RNA-seq tasks? A light discussion would be fine.
2. The types of cell used in constructing DiverseRNA-1.4M is limited to sc-RNA-seq. I don’t get how the bulk RNA-seq data plays role in pretraining step.

---

### Meta-Review · Area_Chair_5Bjk · 2026-02-27

**Recommendation:** Accept (Poster)
**Confidence:** 4

**Metareview:**

This submission has received one very positive review with a "strong accept" and "marginally above acceptance threshold" review.

After reading the reviews, I recommend this paper for "acceptance" and strongly suggest that the authors of this submission consider the suggestions of reviewer "wiir". Please also consider modifying the title of this submission to better represent its content.

---

### Decision · Program_Chairs · 2026-03-03

Accept (Poster)